# A deterministic, c-di-GMP-dependent program ensures the generation of phenotypically similar, symmetric daughter cells during cytokinesis

María Pérez-Burgos[1,4], Marco Herfurth [1,4], Andreas Kaczmarczyk [2], Andrea Harms[1], Katrin Huber[1], Urs Jenal [2], Timo Glatter [3] & Lotte Søgaard-Andersen [1] ✉

Phenotypic heterogeneity in bacteria can result from stochastic processes or deterministic programs. The deterministic programs often involve the versatile second messenger c-di-GMP, and give rise to daughter cells with different c-di-GMP levels by deploying c-di-GMP metabolizing enzymes asymmetrically during cell division. By contrast, less is known about how phenotypic heterogeneity is kept to a minimum. Here, we identify a deterministic c-di-GMP-dependent program that is hardwired into the cell cycle of *Myxococcus xanthus* to minimize phenotypic heterogeneity and guarantee the formation of phenotypically similar daughter cells during division. Cells lacking the diguanylate cyclase DmxA have an aberrant motility behaviour. DmxA is recruited to the cell division site and its activity is switched on during cytokinesis, resulting in a transient increase in the c-di-GMP concentration. During cytokinesis, this c-di-GMP burst ensures the symmetric incorporation and allocation of structural motility proteins and motility regulators at the new cell poles of the two daughters, thereby generating phenotypically similar daughters with correct motility behaviours. Thus, our findings suggest a general c-di-GMP-dependent mechanism for minimizing phenotypic heterogeneity, and demonstrate that bacteria can ensure the formation of dissimilar or similar daughter cells by deploying c-di-GMP metabolizing enzymes to distinct subcellular locations.

In bacteria, motility and its regulation contribute to the colonization of hosts and other habitats, biofilm formation, virulence, and predation[1]. The ubiquitous second messenger cyclic di-GMP (c-di-GMP) is a key regulator of bacterial motility[2–4]. Generally, high c-di-GMP levels inhibit flagella-based swimming motility and stimulate surface adhesion and type IV pili (T4P)-dependent surface motility, thereby promoting surface colonization, biofilm formation, and virulence[2–10]. Several bacterial species that alternate between planktonic and surface-adhered lifestyles harness this duality of c-di-GMP—inhibition of flagella-based motility and stimulation of surface adhesion and T4P-based motility—to establish deterministic programs that are hardwired into the cell cycle to produce phenotypically distinct daughter cells during cell division, thereby generating phenotypic heterogeneity within a population of genetically identical cells[4,11]. Here, we report that

[1]Department of Ecophysiology, Max Planck Institute for Terrestrial Microbiology, Marburg, Germany. [2]Biozentrum, University of Basel, Basel, Switzerland. [3]Core Facility for Mass Spectrometry & Proteomics, Max Planck Institute for Terrestrial Microbiology, Marburg, Germany. [4]These authors contributed equally: María Pérez-Burgos, Marco Herfurth. ✉e-mail: sogaard@mpi-marburg.mpg.de

a c-di-GMP-based, deterministic program is hardwired into the cell cycle of *Myxococcus xanthus* to minimize phenotypic heterogeneity and guarantee the formation of phenotypically similar daughter cells during division.

*M. xanthus* does not have flagella and only translocates on surfaces. For this, the rod-shaped cells use two motility systems, one for T4P-dependent motility and one for gliding[12]. Motility and its regulation are important for the social behaviors of *M. xanthus*, including the formation of spreading colonies in which cells prey on other microbes and spore-filled fruiting bodies in the absence of nutrients[13]. Both motility systems are highly polarized, i.e., the core T4P machine (T4PM) is present at both cell poles[14–17] but only active at one pole at a time, ensuring unipolar T4P assembly[18], and the Agl/Glt machine for gliding only assembles at one pole at a time[19–24]. Because the two machines are active at the same pole, cells translocate unidirectionally across surfaces with a piliated leading and non-piliated lagging cell pole[25–27].

The unipolar assembly of the active motility machines is established by the polarity module. The output of this module is generated by the small GTPase MglA[28,29]. In its GTP-bound active form, MglA localizes to the leading pole[28,29] where it interacts with effectors to stimulate T4P formation and assembly of the Agl/Glt machine[24,26,27,30]. The bipartite guanine nucleotide exchange factor (GEF) RomR/RomX[31] and the bipartite GTPase activating protein (GAP) MglB/RomY[28,29,32] regulate the nucleotide-bound state and, therefore, the localization of MglA[12,24,28,29,31–33]. These four proteins, together with the MglC adapter protein, also localize asymmetrically to the poles[28,29,31,32,34,35].

To regulate their social behaviors, *M. xanthus* cells occasionally reverse their direction of movement[36]. The Frz chemosensory system induces these reversals[36] and acts on the polarity module to induce an inversion of its polarity[33,37], thereby laying the foundation for activating the motility machineries at the new leading pole[12]. Because mutants lacking one or more polarity proteins have an abnormal localization of MglA, such mutants are either non-motile or have an abnormal motility behavior, i.e., hyper-reverse independently of the Frz system or hypo-reverse and are non-responsive to Frz signaling[24,28,29,31,32,35,38].

The c-di-GMP level is determined by the opposing activities of diguanylate cyclases (DGCs), which contain the catalytic GGDEF domain and synthesize c-di-GMP, and phosphodiesterases (PDEs), which degrade c-di-GMP[2,3]. *M. xanthus* encodes 11 GGDEF domain proteins predicted to have DGC activity[39]. The systematic inactivation of the 11 corresponding genes identified DmxA as the only DGC implicated in motility during growth[39], while the DGC DmxB is specifically important for fruiting body formation[40]. The functions of the remaining nine DGCs are not known.

Here, we addressed the mechanism of DmxA. We report that DmxA is recruited to the cell division site, and its DGC activity is switched on late during cytokinesis, resulting in a dramatic but transient increase in the c-di-GMP concentration. The burst in c-di-GMP ensures the equal incorporation and allocation of structural motility proteins and polarity proteins at the new cell poles of the two daughters, thereby generating mirror-symmetric, phenotypically similar daughters with correct motility behavior. Thus, we provide evidence that, during cell division, c-di-GMP guarantees the generation of phenotypically similar offspring.

## Results

### DmxA is a dimeric DGC with a low-affinity I-site

Based on sequence analysis, DmxA has an N-terminal transmembrane domain (TMD) with six α-helices, followed by two GAF domains (named after the protein families cGMP-specific phosphodiesterases, adenylyl cyclases, and FhlA, in which it was first identified), and the catalytic GGDEF domain with the active (A)-site and a c-di-GMP-binding inhibitory (I)-site (Fig. 1a), which is involved in allosteric feedback inhibition of activity in other DGCs[41]. A His$_6$-DmxA variant comprising

the two GAF domains and the GGDEF domain is enzymatically active and binds c-di-GMP in vitro[39]. To understand how the different domains contribute to catalytic activity, we purified five soluble MalE-tagged DmxA variants (Figs. 1b and S1a–c), i.e., variants containing the two GAF domains and the GGDEF domain (MalE-DmxA$^{WT}$), only the two GAF domains (MalE-DmxA$^{GAF×2}$), only the GGDEF domain (MalE-DmxA$^{GGDEF}$), and MalE-DmxA$^{WT}$ variants with substitutions in either the catalytic site (MalE-DmxA$^{E626A}$) or the I-site (MalE-DmxA$^{R615A}$). In size exclusion chromatography (SEC), all variants except MalE-DmxA$^{GGDEF}$ eluted at volumes estimated to correspond to dimers of a globular protein, while the MalE-DmxA$^{GGDEF}$ elution volume would correspond to a monomer of a globular protein (Figs. 1b and S1a, b), indicating that the GAF domain-containing region could be important for dimer formation. Indeed, a high-confidence Alphafold-Multimer structural model supports that DmxA forms a symmetric dimer in which the protomers interact extensively via two α-helices connecting GAF1 to GAF2, and GAF2 to the GGDEF domain and in which the two A-sites are in close proximity and facing each other (Figs. 1c and S1d). Consistently, active DGCs are dimeric, dimerization is mediated by domain(s) outside of the GGDEF domain[42,43], and in solved structures of DGCs, the two A-sites are in close proximity[42–45].

MalE-DmxA$^{WT}$ and MalE-DmxA$^{R615A}$ had DGC activity, while MalE-DmxA$^{E626A}$ and MalE-DmxA$^{GAF×2}$, as expected, did not (Fig. 1d). Monomeric MalE-DmxA$^{GGDEF}$ did not have DGC activity (Fig. 1d), providing additional support that the GAF domain-containing region is important for dimerization and, therefore, DGC activity. In Bio-Layer Interferometry, MalE-DmxA$^{WT}$ and MalE-DmxA$^{E626A}$ bound c-di-GMP, while MalE-DmxA$^{R615A}$, as expected, and MalE-DmxA$^{GAF×2}$ did not; we determined a K$_D$ of 3.5 μM for MalE-DmxA$^{WT}$ (Figs. 1e, f and S1e). C-di-GMP did not significantly inhibit MalE-DmxA$^{R615A}$, while MalE-DmxA$^{WT}$ was inhibited in a cooperative manner ($n_h$ = 1.8), with an inhibitory constant (K$_i$) of ~15 μM (Fig. 1g). This concentration is not only ~10-fold higher than the c-di-GMP concentration ($1.4 \pm 0.5$ μM) previously measured in an unsynchronized population of *M. xanthus* cells[46] but also significantly higher than in other DGCs, e.g., DgcA and PleD of *Caulobacter crescentus* have K$_i$'s of 1 and 6 μM, respectively[41].

We conclude that DmxA has DGC activity and a low-affinity I-site. Moreover, the GAF domain-containing region is important for dimerization, as described for eukaryotic PDEs with an analogous domain architecture[47].

### The ΔdmxA mutant has an aberrant motility behavior with aberrant reversals and a cell polarity defect

A disruption of *dmxA* by a plasmid insertion (Ω*dmxA*) caused reduced T4P-dependent motility[39]. We generated an in-frame deletion mutation of *dmxA* (Δ*dmxA*) to understand the underlying mechanism. In population-based motility assays on 0.5% agar, which is most favorable for T4P-dependent motility, wild-type (WT) generated expanding colonies with the characteristic flares, while the Δ*pilA* negative control, which lacks the major pilin of T4P[48], did not (Fig. 2a). On 1.5% agar, which is most favorable for gliding, WT displayed the characteristic single cells at the colony edge, while the Δ*gltB* negative control, which lacks a component of the Agl/Glt gliding machine[19], did not (Fig. 2a). The Δ*dmxA* mutant displayed significantly reduced colony expansion under both conditions and these defects were corrected upon complementation (Fig. 2a). To corroborate that both motility systems were affected by lack of DmxA, we generated Δ*dmxA* mutants additionally lacking either GltB or PilA. These double mutants can only move by means of T4P or gliding, respectively. The Δ*dmxA*Δ*gltB* double mutant, had significantly reduced colony expansion on 0.5% agar compared to the Δ*gltB* mutant (Fig. 2a). Thus, Δ*dmxA* causes a defect in T4P-dependent motility. The Δ*dmxA*Δ*pilA* double mutant had single cells at the colony edge but a more significant expansion defect than the Δ*pilA* mutant (Fig. 2a). Thus, the Δ*dmxA* mutation also causes a gliding defect. We previously reported that the Ω*dmxA* mutant only had a

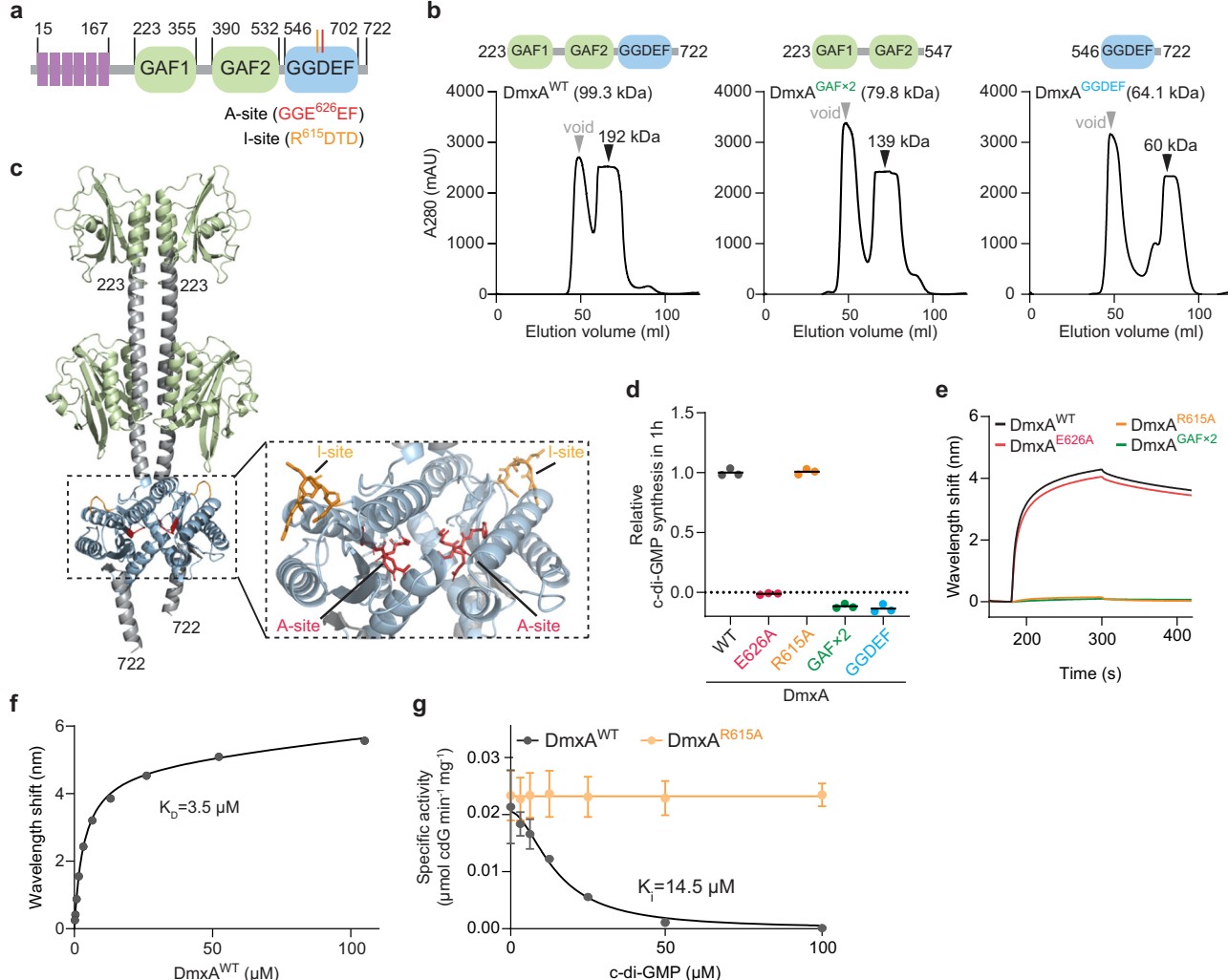

**Fig. 1 | DmxA has DGC activity and a low-affinity I-site. a** Domain architecture of DmxA. **b** SEC of MalE-DmxA variants. Domain architecture of DmxA variants is shown above chromatograms. Gray arrowheads indicate void volume, and black arrowheads elution volume with the corresponding calculated molecular weight. **c** AlphaFold-multimer structural model of dimeric DmxA. The transmembrane helices were removed before generating the model, residue numbers are indicated. Inset, side chains of residues in A- and I-sites are indicated in red and orange, respectively, in ball-and-stick representation. **d** In vitro DGC assay of the indicated MalE-DmxA variants. The relative amount of c-di-GMP synthesized after 1 h is determined by measuring released inorganic pyrophosphate. Measurements from three technical replicates are shown relative to the mean (black lines) of MalE-DmxA$^{WT}$. **e** Bio-layer interferometric analysis of c-di-GMP binding by MalE-DmxA

variants. Streptavidin-coated sensors were loaded with 500 nM biotinylated c-di-GMP and probed with 10 μM of the indicated proteins. The interaction kinetics were followed by monitoring the wavelength shift during the association and dissociation of the analyte. **f** Determination of $K_D$ of MalE-DmxA for c-di-GMP. The plot shows the equilibrium levels measured at the indicated MalE-DmxA$^{WT}$ concentrations (see also Fig. S1e). The data were fitted to a non-cooperative one-site specific-binding model. **g** Determination of $K_i$ of MalE-DmxA for c-di-GMP. Inhibition of the specific activity of DmxA$^{WT}$ and DmxA$^{R615A}$ DGC activity over time was measured as in (**d**) in the presence of different c-di-GMP concentrations. Points and error bars represent the mean ± standard deviation (SD) calculated from three biological replicates. The data were fitted to an inhibition model with a variable slope. Source data are provided as a Source Data file.

defect in T4P-dependent motility[39]. However, in those experiments, we exclusively focused on the presence or absence of single cells at the colony edge and not overall colony expansion.

Generally, mutants with defects in both motility systems have defects in the regulation of the reversal frequency[24,28,29,31–33,35,38]. Because the lack of DmxA affected both motility systems, we hypothesized that the reversal frequency of the Δ*dmxA* mutant could be affected. To test this idea, we assayed single-cell motility. We observed that Δ*dmxA* cells had the same speed as WT (Fig. 2b, c). However, the Δ*dmxA* cells reversed aberrantly compared to WT, ranging from cells that did not reverse to cells that reversed up to eight times during the experiment (Fig. 2b, c). Using the median absolute deviation (MAD) as a robust measure of variation, we found that the distribution of reversal frequencies in the Δ*dmxA* mutant was significantly broader

than in WT (Fig. 2b, c). The reversal defect was corrected upon complementation.

Δ*frzE* cells, which lack the FrzE kinase essential for Frz-induced reversals[49], as expected, did not reverse (Fig. 2b, c). However, many Δ*dmxA*Δ*frzE* cells still reversed. Mutants with Frz-independent reversals generally lack polar MglB/RomY GAP activity, causing MglA to localize to both cell poles[24,28,29,32]. Interestingly, a comparison of the Δ*dmxA* and Δ*mglB* mutants demonstrated that the Δ*dmxA* mutant had a much broader distribution of reversal frequencies than the Δ*mglB* mutant, in which essentially all cells reverse or hyper-reverse (Fig. 2b, c).

Because lack of GAP activity results in T4P formation at both cell poles in many cells[27], we determined the piliation pattern of Δ*dmxA* cells using transmission electron microscopy. About 82% of WT cells

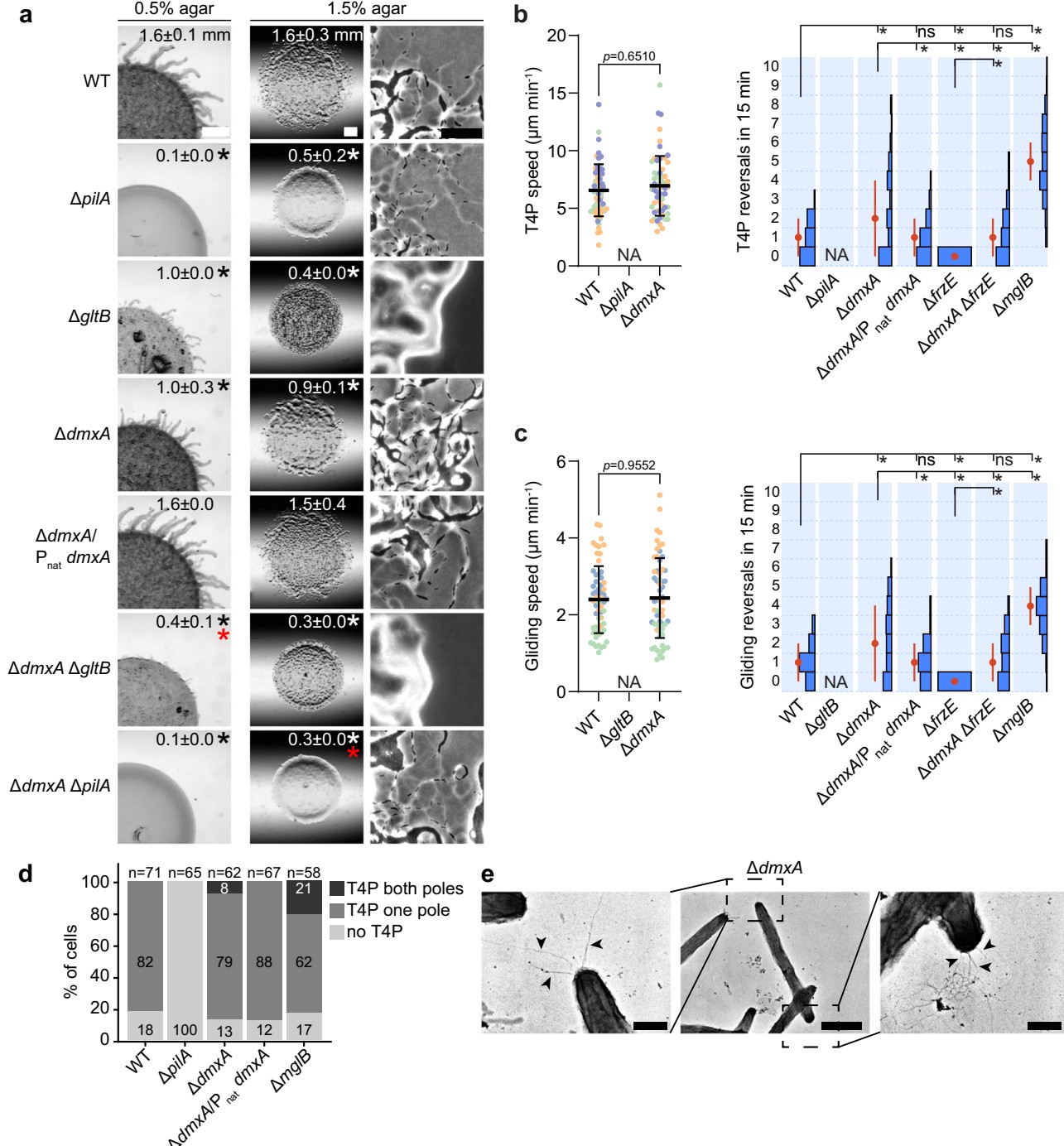

**Fig. 2 | The ΔdmxA mutant has an aberrant motility behavior with aberrant reversals and a cell polarity defect. a** Population-based motility assay. T4P-dependent motility and gliding were analysed on 0.5 and 1.5% agar, respectively. Numbers indicate the colony expansion in 24 h as mean ± SD ($n = 3$ biological replicates). In the complementation strain, *dmxA* was expressed from its native promoter from a plasmid integrated into a single copy at the Mx8 *attB* site. Black, white, and red asterisks indicate $p < 0.05$, two-sided Student's *t*-test against WT (black, white) or the Δ*pilA* or Δ*gltB* controls (red). Scale bars, 1 mm (left), 1 mm (middle), 50 μm (right). **b**, **c** Single cell-based motility assays. T4P-dependent motility was measured for cells on a polystyrene surface covered with 1% methyl-cellulose (**b**) and gliding on 1.5% agar supplemented with 0.5% CTT (**c**). Cells were imaged for 15 min with 30 s intervals. Speed ($n = 60$ from three biological replicates with each 20 cells indicated in different colors) and number of reversals ($n = 90$

from three biological replicates with each 30 cells). Only cells moving during the entire recording interval were included. For speed, error bars represent mean ± SD calculated from all data points. Reversals are represented as histograms based on all three replicates, error bars indicate the median±median absolute deviation (MAD). NA not applicable because cells are non-motile. \**p* < 0.05, ns not significant; statistical tests: speed, two-sided Mann–Whitney test, reversals, one-way ANOVA multiple comparison test, and Fisher's LSD test. **d** Quantification of T4P localization based on transmission electron microscopy. Total number of cells from at least three biological replicates indicated above. **e** Transmission electron microscopy of a Δ*dmxA* cell, with T4P at both poles. T4P are indicated by black arrowheads. The image in the middle shows the complete cell and the images on the left and right show the two poles. Scale bars, 0.5 μm (left and right), 2 μm (middle). Source data, including exact *p* values, are provided as a Source Data file.

were unipolarly piliated and the remaining cells unpiliated, while 21% of Δ*mglB* cells were bipolarly and 62% unipolarly piliated (Figs. 2d and S2a). Importantly, 8% of Δ*dmxA* cells were bipolarly and 79% unipolarly piliated (Figs. 2d, e and S2a). Consistently, in a T4P shear-off assay, in which T4P are sheared off the cell surface followed by quantification of PilA levels by immunoblotting, PilA was present in the sheared fraction from the Δ*dmxA* mutant at a slightly but significantly higher level than in WT (Fig. S2b). Also, and in agreement with T4P formation and production of the secreted polysaccharide EPS stimulating each other[50], the Δ*dmxA* mutant produced slightly more EPS than WT (Fig. S2c), as previously reported for the Ω*dmxA* mutant[39]. Finally, the structural proteins of the two motility machines, the Frz proteins, and the proteins of the polarity module accumulated at the same level in WT and the Δ*dmxA* mutant (Fig. S2d–i), supporting that changes in protein accumulation are not responsible for the aberrant motility behavior.

We conclude that cells lacking DmxA have motility defects caused by aberrant reversals and have an unusually broad distribution of reversal frequencies. Moreover, the Δ*dmxA* mutant has a cell polarity defect, and the underlying mechanism differs from that of a mutant lacking MglB/RomY GAP activity.

## DmxA is recruited to the division site by the divisome late during cytokinesis

To understand how DmxA impacts motility behavior and cell polarity, we determined the localization of an active (see below) DmxA-mVenus fusion expressed from the native site. In fluorescence microscopy snapshots, we found that DmxA-mVenus localized at mid-cell in ~5% of cells, while the remaining cells had a speckle-like pattern along the cell body (Fig. 3a). Importantly, in all the cells with a mid-cell cluster, the cluster co-localized with a cell division constriction. Still, not all constricting cells had a DmxA-mVenus cluster, suggesting that DmxA-mVenus is recruited to the division site late during cytokinesis. Indeed, in time-lapse microscopy recordings in which cells were followed over the cell cycle, we observed that (i) DmxA-mVenus formed a cluster at the division site in all division events but only after initiation of cytokinesis ($n = 100$), (ii) the cluster appeared at the division site $20 \pm 15$ min before completion of cytokinesis, and (iii) disintegrated upon completion of cytokinesis. Consistently, the mean cluster lifetime was $20 \pm 16$ min (Fig. 3b). The ~5 h doubling time of *M. xanthus* and this cluster lifetime agree well with the percentage of cells with a mid-cell cluster in an unsynchronized cell population.

To begin to understand the mechanism underlying the cell cycle-dependent DmxA localization, we determined the steady-state levels of DmxA-mVenus in a mixed population of cells based on snapshot images. To this end, we measured the total cellular DmxA-mVenus fluorescence ($n = 1329$) and normalized the fluorescence intensities for cell area to obtain the normalized cellular DmxA-mVenus fluorescence and used this metric as a measure of the normalized DmxA-mVenus concentration as previously described[51,52]. These analyses showed that all cells accumulated DmxA-mVenus and that the cellular DmxA-mVenus concentration followed a unimodal distribution (Fig. S3a). Consistently, based on time-lapse microscopy, we found that the normalized fluorescence intensities of DmxA-mVenus remained constant before, during, and after cytokinesis (Fig. S3b). Based on these analyses, we infer that the steady-state level of DmxA-mVenus is constant over the cell cycle, strongly suggesting that localization of DmxA-mVenus to the division site, but not accumulation of DmxA, is cell cycle regulated.

DmxA is encoded in an operon with the FtsB divisome protein, and this genetic organization is conserved in related species (Fig. S3c–f). However, Δ*dmxA* cells had neither a growth (Fig. S3g) nor a cell length defect (Fig. S3h). Therefore, to test whether DmxA is recruited to the division site by the divisome, we first treated cells with cephalexin, which blocks cytokinesis after initiation of constriction in *M. xanthus*[53]. In cells treated with cephalexin for one generation, the frequency of cells with a constriction and a DmxA-mVenus cluster at mid-cell had significantly increased (Fig. 3a). DmxA-mVenus accumulated at the same level in cephalexin-treated and untreated cells (Fig. S4a), hinting that DmxA-mVenus synthesis is not cell cycle-regulated. Next, we depleted *dmxA-mVenus* expressing cells of the essential cell division protein FtsZ[53] using a strain in which the only copy of *ftsZ* was expressed from a vanillate-inducible promoter (P_{van}). In the presence of vanillate, DmxA-mVenus localized at constrictions in 3% of cells (Fig. 3c). At 10 h of depletion, FtsZ was undetectable by immunoblotting (Fig. S4b), and cells had neither constrictions nor DmxA-mVenus clusters at mid-cell (Fig. 3c) despite the protein accumulating at a slightly higher level than in untreated cells (Fig. S4b). Finally, we observed in proximity labeling experiments using strains expressing either DmxA or sfGFP fused to the promiscuous biotin ligase miniTurbo (Fig. S4c) that the cell division protein FtsK, which also localizes to the division site in *M. xanthus*[54], was significantly enriched in the DmxA-miniTurbo samples (Fig. 3d and Table S1). Equally, in the reciprocal experiment using an FtsK-miniTurbo construct (Fig. S4c and Table S2), DmxA was significantly enriched (Fig. 3e).

Altogether, these observations support that DmxA accumulation is not cell cycle-regulated and that DmxA interacts with protein(s) of the divisome, and is thereby recruited to the division site late during cytokinesis.

## DmxA function depends on DGC activity and DmxA is recruited to the division site by the TMD

To determine whether DGC activity and which domains contribute to DmxA function and localization, we replaced *dmxA* with *mVenus*-fused versions of full-length *dmxA*^WT, *dmxA*^E626A and *dmxA*^R615A as well as with the truncated variants *dmxA*^ΔGAF×2, *dmxA*^ΔTMD and *dmxA*^TMD (Fig. 3f). By immunoblot analysis, all variants accumulated at similar levels except DmxA^ΔGAF×2-mVenus and *dmxA*^TMD-mVenus, which accumulated at slightly lower levels (Fig. S5a). Among these variants, only DmxA^WT-mVenus and DmxA^R615A-mVenus supported WT motility (Figs. 3g and S5b). DmxA-mVenus localization to the division site was independent of DGC activity, the I-site and the two GAF domains (Fig. 3g). By contrast, the TMD was not only essential but also sufficient for DmxA-mVenus localization to the division site (Fig. 3g). Thus, DmxA function depends on DGC activity but not on c-di-GMP binding to the I-site, and DmxA is recruited to the division site by the TMD.

## DmxA DGC activity is activated upon recruitment to the division site

Based on the DmxA localization pattern, we hypothesized that DmxA activity is cell cycle-regulated and explicitly switched on late during cytokinesis. To test this idea, we used the genetically encoded c-di-GMP biosensor cdGreen2[55], for which binding of c-di-GMP results in conformational changes leading to increased green fluorescence, thus allowing real-time measurements of c-di-GMP levels at single-cell resolution over the entire cell cycle. For normalization of the cdGreen2 signal, *cdGreen2* was co-expressed with a gene encoding the fluorescent protein mScarlet-I[55].

In snapshots of WT cells, we observed significant cell-to-cell variability of the cdGreen2 signal, while the mScarlet-I signal varied much less (Figs. 4a and S6a, b). Intriguingly, this cell-to-cell variability was clearly bimodal and only long cells with a constriction and some very short cells had a very high cdGreen2 signal (Figs. 4a and S6a). To focus on DmxA, we generated a mutant lacking 10 of the 11 predicted DGCs in *M. xanthus*, leaving DmxA as the only DGC (henceforth the Δ10 mutant). In snapshots of the Δ10 mutant, DmxA-mVenus localized to the cell division site as in WT (Fig. S6c). Remarkably, the cdGreen2-signal was even more clearly bimodal in this strain, i.e., high in long cells with a constriction and in some very short cells (Figs. 4a and S6b).

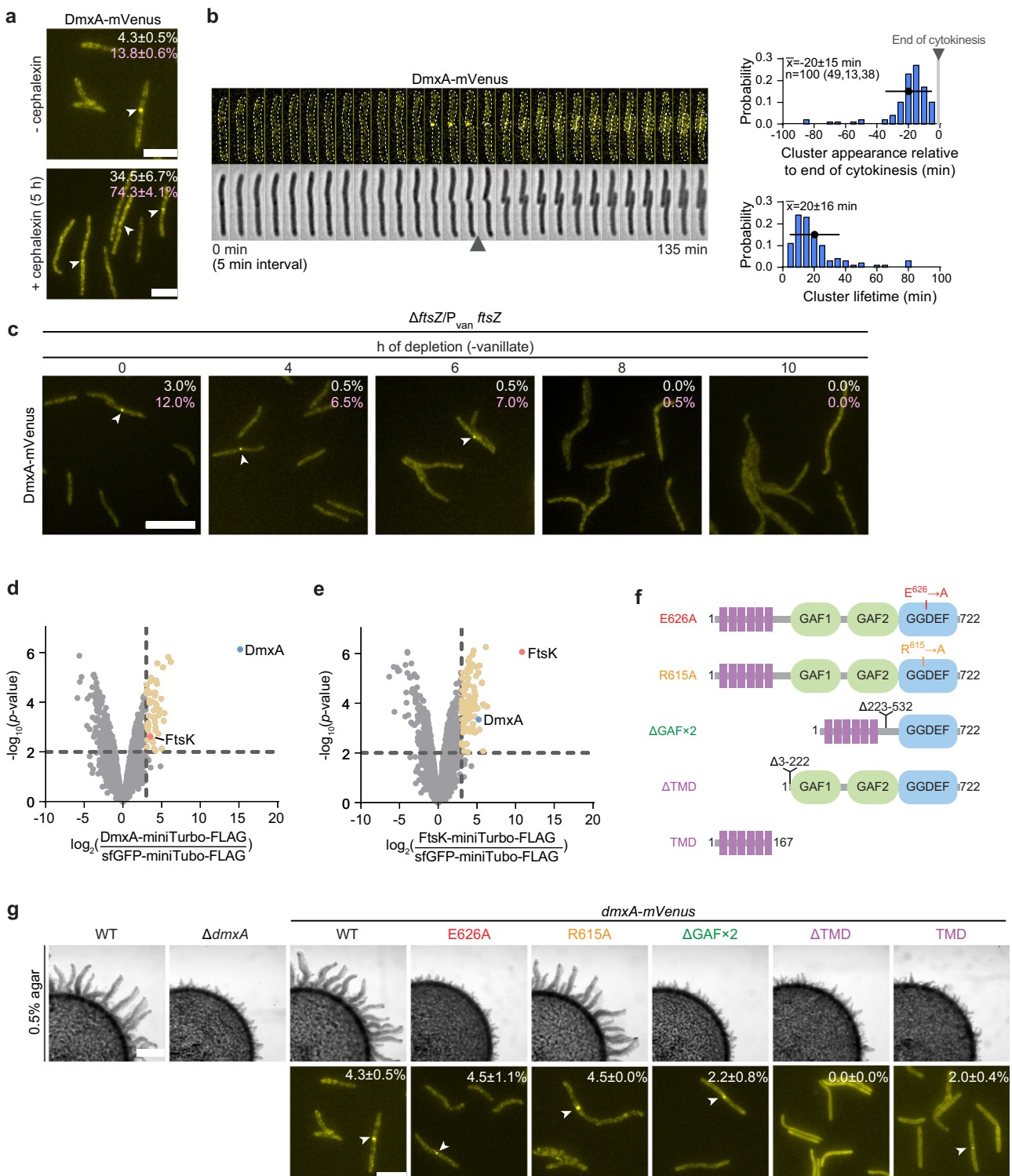

By contrast, all Δ*dmxA* cells, i.e., cells only lacking DmxA but retaining the remaining ten DGCs, had the same low cdGreen2 background signal (Figs. 4a and S6b).

Next, we used time-lapse microscopy to follow the cdGreen2 signal dynamics over time. The analysis of Δ10 (*n* = 130) and WT (*n* = 73) cells revealed that the cdGreen2 signal specifically increased dramatically in all cells shortly before completion of cytokinesis and then decreased rapidly in the two daughters, which had equal levels of cdGreen2 fluorescence (Figs. 4b and S6d). By contrast, Δ*dmxA* cells (*n* = 67) completely lacked the transient increase in c-di-GMP (Fig. 4c). The increase in c-di-GMP initiated 22 ± 12 min before

completion of cytokinesis, and remained high for 19 ± 9 min (Fig. 4b). Remarkably, DmxA-mVenus localizes to the division site with a similar timing (Fig. 3b). Moreover, in cephalexin-treated cells, the DmxA-mVenus clusters persisted longer at the division site and then eventually disintegrated (Fig. S6e). Notably, the high cdGreen2 signal also persisted longer in cephalexin-treated cells and eventually also vanished (Fig. S6f). We conclude that DmxA is sufficient and required for the transient c-di-GMP burst during cytokinesis.

The fully active DmxA[R615A] I-site variant also supported the burst in c-di-GMP, while the non-complementing DmxA[ΔTMD] and DmxA[ΔGAF×2] variants did not (Figs. 4a and S6b). Importantly, DmxA[ΔTMD]-mVenus is

**Fig. 3 | DmxA is recruited to the division site during cytokinesis by the divisome and its function depends on DGC activity. a** Localization of DmxA-mVenus in the presence and absence of cephalexin. The mean ± SD percentage of cells with a DmxA-mVenus cluster (white) or a constriction (pink) is indicated; $n = 600$ from three biological replicates with each 200 cells. Scale bars, 5 μm. **b** DmxA-mVenus localization during the cell cycle. Left panel, epifluorescence and phase-contrast images from time-lapse microscopy of a representative cell expressing DmxA-mVenus. Images recorded every 5 min; arrowhead indicates completion of cytokinesis (defined as the first frame at which daughters were clearly separated); right panels, histograms of the timing of cluster appearance relative to the completion of cytokinesis and lifetime of DmxA-mVenus clusters ($n = 100$ division events from three biological replicates, number of events per replicate in brackets). The first time point after completion of cytokinesis is defined as $t = 0$ and indicated by a gray vertical bar. Error bar, mean ± SD. **c** Localization of DmxA-mVenus during FtsZ depletion. Cells were grown in the presence of 10 μM vanillate before the depletion. The percentage of cells with a cluster (white) or a constriction (pink) are indicated ($n = 200$ from one biological replicate). White arrowheads indicate DmxA-mVenus clusters. Scale bar, 10 μm. **d, e** Proximity labeling using DmxA-miniTurbo-FLAG or FtsK-miniTurbo-FLAG as baits compared to sfGFP-miniTurbo-FLAG. Volcano plots show proteins enriched by DmxA-miniTurbo-FLAG (**d**) and FtsK-miniTurbo-FLAG

(**e**). DmxA-miniTurbo-FLAG and FtsK-miniTurbo-FLAG were expressed from the *pilA* promoter, and sfGFP-miniTurbo-FLAG from the P$_{van}$ in the presence of 100 μM vanillate added 18 h before the addition of 100 μM biotin and cephalexin for 4 h. Samples from three biological replicates were analysed. X-axis, log$_2$-fold enrichment in experimental samples compared to sfGFP-miniTurbo-FLAG (negative control) calculated based on normalized intensities. Y-axis, -log$_{10}$ of $p$ value. Significantly enriched proteins in the experimental samples (log$_2$ ratio ≥3; $p$ value ≤ 0.01 (−log$_{10}$ ≥ 2.0) are indicated by the stippled lines. DmxA and FtsK are shown in blue and red, respectively, and other enriched proteins are in yellow. Enriched proteins other than FtsK and DmxA are listed in Tables S1 and S2. $p$ values were calculated using eBayes moderated $t$-statistics[109] and are listed in the Tables S1 and S2. **f, g** Analysis of DmxA-mVenus variants. Domain architecture of variants (**f**) and population-based motility assay for T4P-dependent motility ((**g**), upper panels) and localization ((**g**), lower panels). Motility was analysed as in Fig. 2a. In lower panels, the mean ± SD percentage of cells with cluster at mid-cell is indicated. White arrowheads indicate clusters. In (**g**), the data for WT are the same as in (**a**), and were included for presentation purposes. For all strains, $n = 600$ from three biological replicates with each 200 cells. Scale bars, 1 mm (upper panels), 5 μm (lower panels). Source data are provided as a Source Data file.

similar to MalE-DmxA$^{WT}$, which has DGC activity in vitro (Fig. 1d), but DmxA$^{ΔTMD}$-mVenus does not localize to the division site (Fig. 4a), arguing that DmxA DGC activity in vivo is explicitly activated upon its recruitment to the division site. Furthermore, DmxA$^{ΔGAFx2}$ localizes to mid-cell but lacks the region for DmxA dimerization, arguing that this region also in vivo mediates dimerization and, thus, DGC activity.

Finally, we observed no significant difference in the c-di-GMP levels in unsynchronized populations of WT and Δ*dmxA* cells (Fig. S6g), corroborating that DmxA only displays DGC activity for a brief period (~20 min) of the 5 h cell cycle. We note that we previously reported that the Ω*dmxA* mutant had a 1.5-fold higher level of c-di-GMP than WT[39]. We suggest that this difference is caused by using two different *dmxA* mutants.

To resolve whether the increase in c-di-GMP initiates before or after the separation of the cytoplasm of the two daughters, we performed fluorescence recovery after photobleaching (FRAP) experiments. In these experiments, we bleached the mScarlet-I signal in one-half of predivisional cells ($n = 22$) and, in parallel, we imaged the cdGreen2 signal to verify that the cells contained a high c-di-GMP level. The bleaching event caused a decrease in the mScarlet-I fluorescence signal in both cell halves in 50% of cells, suggesting that the cytoplasm is still connected when DmxA is activated, and only affected the signal in the bleached half in the remaining 50% (Fig. 4d).

We conclude that DmxA causes the transient c-di-GMP burst during cytokinesis and that DGC activity is explicitly activated upon recruitment to the division site and before the cytoplasm of the daughters is separated.

## DmxA is essential for the symmetric incorporation and allocation of the core T4PM at the nascent and new cell poles

We next sought to establish the link between DmxA DGC activity during cytokinesis, motility, and cell polarity. Because the T4PM core is present at both cell poles, is symmetrically incorporated at the nascent and new poles during and immediately after completion of cytokinesis[16,56], and DmxA is active during cytokinesis, we initially focused on the polar assembly of the core T4PM. This assembly starts with forming the PilQ secretin in the outer membrane (OM), progresses inward, and culminates with the incorporation of cytoplasmic PilM[16] (Fig. S2f). The PilB and PilT ATPases bind to the cytoplasmic base of the core T4PM at the leading pole in a mutually exclusive manner to stimulate T4P extension and retraction, respectively[15,57,58] (Fig. S2f). Note that PilT localizes either asymmetrically or symmetrically to both poles in most cells[27,59], while PilB localizes bipolar asymmetrically or unipolarly with the large cluster at the leading pole[15,27,59].

First, we performed time-lapse microscopy using PilQ-sfGFP as a readout for assembly of the core T4PM. As reported, in WT, PilQ-sfGFP incorporation was initiated during cytokinesis and, upon completion of cytokinesis, continued symmetrically at the new poles in the two daughters ($n = 28$) (Fig. 5a), giving rise to mirror-symmetric daughters. Strikingly, in the absence of DmxA, PilQ-sfGFP incorporation was not only significantly delayed but also asymmetric in the two daughters ($n = 31$), resulting in asymmetric daughter pairs and with many cells having PilQ-sfGFP clusters of very different intensities or even only a single, unipolar PilQ-sfGFP cluster at the old pole (Fig. 5a). Moreover, this defect in PilQ-sfGFP polarity was typically not fully corrected before the subsequent cell division (Fig. 5a), resulting in propagation of the polarity defect.

Consistent with the faulty polar PilQ incorporation, we observed in quantitative analyses of snapshot microscopy images that PilQ-sfGFP, mCherry-PilM, and mCherry-PilT localization in the Δ*dmxA* mutant was significantly shifted toward unipolar and, thus, less symmetric than in WT (Fig. 5b). The shifts toward asymmetry were primarily caused by many Δ*dmxA* cells having no or a strongly decreased fluorescence signal at the pole with the lowest fluorescence (Fig. 5b). Similarly, PilB-mCherry was significantly more unipolar in Δ*dmxA* cells than in WT (Fig. 5b). Thus, in the absence of DmxA, all tested T4PM proteins localize significantly more asymmetrically than in WT.

We conclude that DmxA and the burst in c-di-GMP are essential for the symmetric incorporation and allocation of PilQ during and after cytokinesis, thereby generating mirror-symmetric daughters. Moreover, we infer that the faulty polar PilQ incorporation and allocation contribute to the more asymmetric localization of the other tested T4PM proteins in the Δ*dmxA* mutant.

## DmxA is essential for the symmetric allocation of the polarity proteins to the daughters during cytokinesis

We predicted that the proteins of the polarity module would also be symmetrically allocated during cytokinesis to generate mirror-symmetry of these proteins in the daughters. Briefly, RomR alone localizes polarly in the absence of the remaining five polarity proteins and brings about polar localization of these proteins[35,51]. Moreover, RomX and MglC localization follows that of RomR[31,35] and RomY the highest concentration of MglB[32] (Fig. S2d). Therefore, we used RomR and MglB as well as MglA, which generates the output of the polarity module, as readouts for the localization of the proteins of the polarity module.

First, to test our prediction, we performed time-lapse microscopy of WT expressing RomR-mCherry. In WT ($n = 27$), these analyses

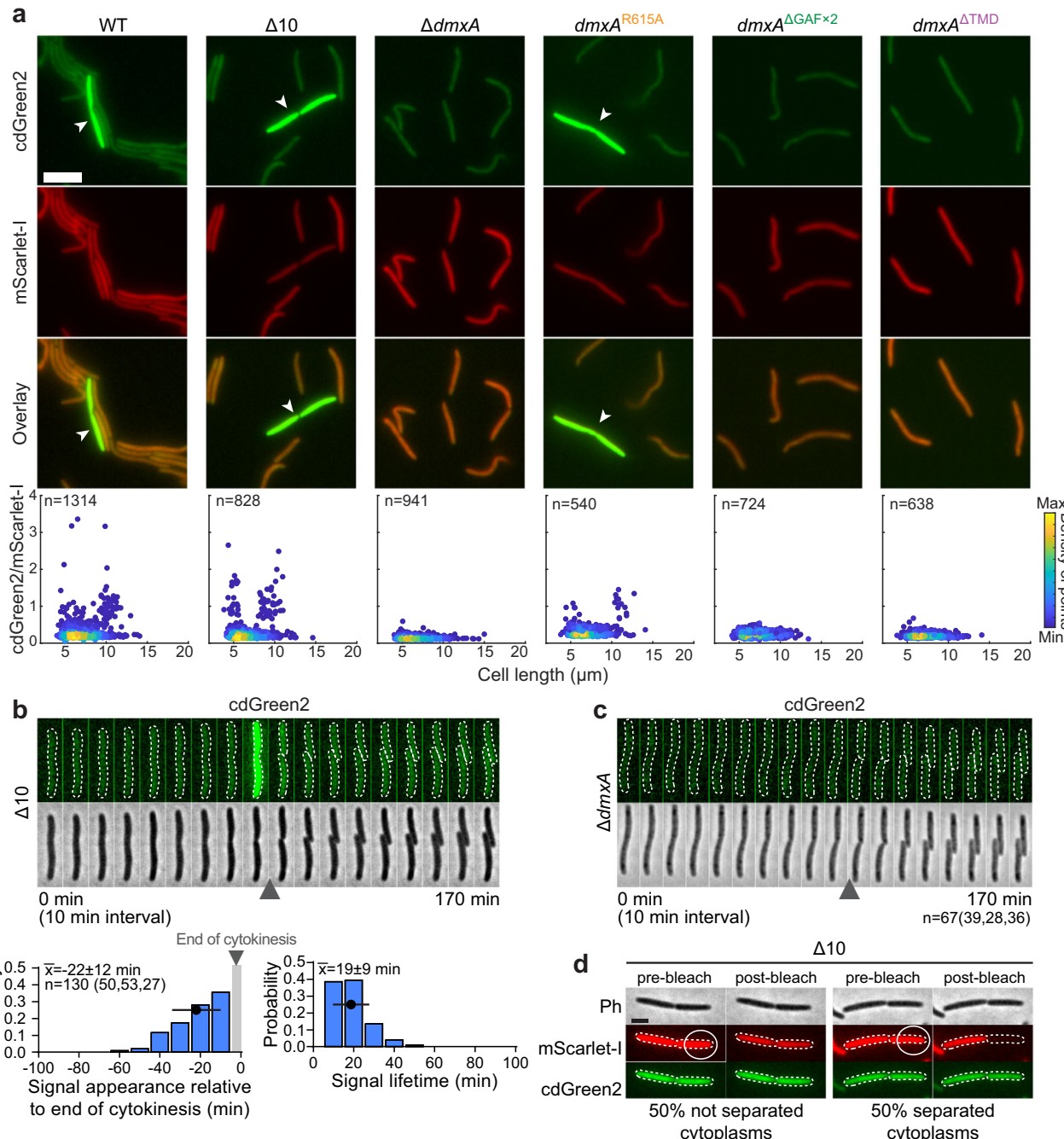

**Fig. 4 | DmxA DGC activity is switched on upon recruitment to the division site.**
**a** Analysis of cdGreen2 and mScarlet-I fluorescence in the indicated strains. Upper panels, epifluorescence snapshot images of representative cells expressing cdGreen2 and mScarlet-I. White arrowheads indicate cells with high cdGreen2 fluorescence. The *cdGreen2-mScarlet-I* operon was expressed from the constitutively active *pilA* promoter. Scale bar, 5 µm. Lower panels, scatter plots of the cdGreen2/mScarlet-I fluorescence ratio of each cell relative to its cell length. Colors indicate the density of points according to the scale on the right. Total number of cells from one biological replicate indicated. **b**, **c** cdGreen2 fluorescence in representative Δ10 (**b**) and Δ*dmxA* (**c**) cells during the cell cycle. Upper panel, epifluorescence and phase-contrast images from time-lapse microscopy of representative cells. Images were recorded every 10 min; arrowheads indicate completion of cytokinesis (defined as the first frame in which daughters were

clearly separated). In (**b**), lower panels, histograms of the timing of appearance of the cdGreen2 fluorescence signal relative to completion of cytokinesis and lifetime of the high cdGreen2 fluorescence signal (*n* = 130 division events from three biological replicates, number of events per replicate in brackets). Error bars, mean ± SD. The first time point after completion of cytokinesis is defined as *t* = 0 and indicated by a gray vertical bar. In (**c**), *n* = 67 division events from three biological replicates, number of events per replicate in brackets. **d** FRAP experiment on predivisional Δ10 cells expressing cdGreen2 and mScarlet-I. The mScarlet-I signal of one half of a cell was bleached in the region indicated by a white circle. Post-bleached images were recorded 2 s after the bleaching event. All predivisional cells analysed had a high cdGreen2 fluorescent signal. *n* = 22 from one biological replicate. Scale bar, 2 µm. Source data are provided as a Source Data file.

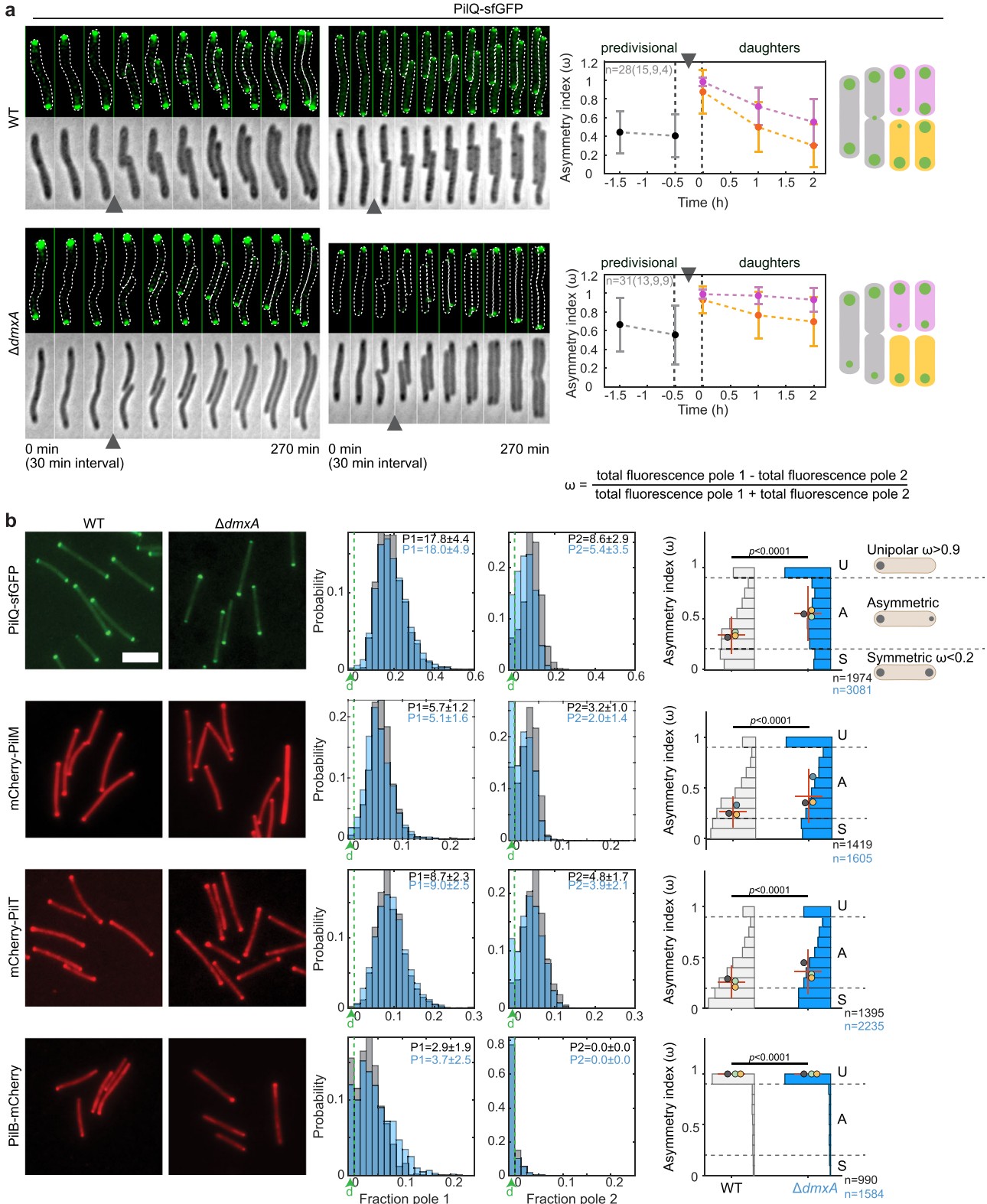

$$\omega = \frac{\text{total fluorescence pole 1 - total fluorescence pole 2}}{\text{total fluorescence pole 1 + total fluorescence pole 2}}$$

revealed a precise order of events in which, late during cytokinesis, RomR-mCherry was released from the old poles, then localized to the division site, and, upon completion of cytokinesis, was symmetrically allocated to the two daughters (Fig. 6a), giving rise to mirror-symmetric daughters. Remarkably, in the absence of DmxA, we observed very different patterns (*n* = 22). During most division events, RomR-mCherry was released from the old poles, however, it was either not recruited at the division site but instead switched to the opposite

pole (Fig. 6a, right), or if it localized to the division site, it was asymmetrically allocated to the daughters (Fig. 6a, left). Consequently, the daughters of a division event contained different amounts of RomR-mCherry, were not mirror-symmetric, had polar clusters of very different intensities or even only a single RomR-mCherry cluster at the old pole, or only had diffused localization of RomR-mCherry. The defects in RomR-mCherry polarity were typically also not fully corrected before the subsequent cell division (Fig. 6a).

**Fig. 5 | DmxA is essential for the symmetric incorporation and allocation of the core T4PM proteins at the nascent and new cell poles. a** PilQ-sfGFP localization during the cell cycle. Left panels, epifluorescence and phase-contrast images from time-lapse microscopy of representative WT and Δ*dmxA* cells. Images were recorded every 30 min; arrowheads indicate completion of cytokinesis. Right panels, quantification of PilQ-sfGFP localization pattern based on time-lapse microscopy. PilQ-sfGFP polar fluorescence was analysed before and after division in the pre-divisional cell (gray) and the daughters (yellow and pink) and the asymmetry index (ω) calculated as indicated; the pole with the highest fluorescence is defined as Pole 1. Daughter cells are sorted based on their ω with the most symmetric in yellow and the other in pink. Line and error bars, mean ± SD from three biological replicates, number of cell division events per replicate in brackets. Schematics show dominant localization patterns of PilQ-sfGFP in WT and Δ*dmxA* cells during the cell cycle. **b** Localization of PilQ-sfGFP, mCherry-PilM, mCherry-PilT and PilB-mCherry in WT and Δ*dmxA* cells. Left panels, representative snapshot images. Scale bar, 5 μm.

Middle panels, histograms of the distribution of the fraction of total cellular fluorescence in polar clusters at pole 1 and pole 2 in WT (gray) and Δ*dmxA* (blue) cells. The pole with the highest fluorescence is defined as Pole 1. Numbers in the upper right corners indicate the median±MAD fluorescence signal at pole 1 (P1) and pole 2 (P2). The fraction of cells with no polar signal(s) is indicated in the leftmost column labeled d (for diffused) in green; cells in which no polar cluster was detected at pole 1 also do not have a signal at pole 2. Right panels, histograms of the distribution of ω. Localization patterns are binned as unipolar, asymmetric, and symmetric from the ω values as indicated; cells in which no polar signal was detected were not considered in the analysis. Error bars indicate median ± MAD. Differently colored circles indicate the median of each of the three biological replicates. The total number of analysed cells is indicated below. Samples were analysed using the two-sided Mann–Whitney test. Source data are provided as a Source Data file.

Consistent with the faulty polar RomR allocation to the daughters in the absence of DmxA, we observed in quantitative analyses of snapshot microscopy images that RomR-mCherry, MglB-mVenus, and MglA-mVenus in the Δ*dmxA* mutant localized in highly aberrant patterns and had largely lost their defined polar asymmetry in individual cells, displaying much broader variations in asymmetry values compared to WT (Fig. 6b). The aberrant RomR-mCherry and MglB-mVenus asymmetry resulted from the much broader variations in the fluorescence signals at both poles (Fig. 6b); accordingly, MglA polar localization was significantly reduced in many cells (Fig. 6b).

We conclude that RomR polarity is reset during cytokinesis in WT and that DmxA together with the burst in c-di-GMP are essential for this reset and the symmetric allocation of RomR to the two daughters. Moreover, we infer that the highly aberrant localization of MglB and MglA in the Δ*dmxA* mutant results from the faulty RomR allocation during cytokinesis.

Finally, we aimed to establish the link between the aberrant localization of the T4PM and the polarity proteins to the aberrant T4P-dependent motility behavior of the Δ*dmxA* cells. Because the polar signals of PilB-mCherry and MglA-mVenus are low, it is technically difficult to follow these fusions in time-lapse recordings. Therefore, we followed the localization of the MglA-GTP effector SgmX-mVenus, which is recruited to the leading pole by MglA, and then recruits the PilB ATPase to stimulate T4P extension[26,27]. In WT, SgmX-mVenus localized with a large cluster at the leading pole in 100% of cells (*n* = 48) and switched polarity during reversals (Fig. 6c). In the Δ*dmxA* mutant (*n* = 84), SgmX-mVenus localized as in WT in ~75% of cells but in the remaining ~25%, SgmX-mVenus localized aberrantly with either a bipolar pattern and/or more unstably at the leading pole, i.e., the intensity at the leading pole would shortly decrease, and this was occasionally accompanied by a brief increase in fluorescence at the opposite pole (Fig. 6c). Importantly, many of these ~25% of cells hyper-reversed (Fig. 6c).

Altogether, these observations support that the aberrant localization of the T4PM and the polarity proteins caused by lack of DmxA results in motility defects with aberrant reversals.

## Discussion

Here, we describe a c-di-GMP-dependent program that is hardwired into the *M. xanthus* cell cycle and guarantees the formation of mirror-symmetric, phenotypically similar daughter cells. Specifically, the DGC DmxA is explicitly recruited to the division site late during cytokinesis, and released upon completion of cytokinesis. During this brief period of the cell cycle, its DGC activity is switched on, resulting in a dramatic but transient increase in the c-di-GMP concentration. This c-di-GMP burst, in turn, ensures the equal and symmetric allocation of core T4PM proteins and polarity proteins to the two daughters. In the absence of DmxA, the daughters inherit unequal amounts of these

proteins causing aberrant T4PM localization and cell polarity and, consequently, aberrant motility behavior. Thus, *M. xanthus* harnesses DmxA and c-di-GMP to ensure the generation of mirror-symmetric, phenotypically similar daughters in each cell division event.

DmxA recruitment to the division site late during cytokinesis depends on the TMD of DmxA and the divisome, suggesting that the TMD interacts with the divisome. Indeed, using proximity labeling, we identified the transmembrane divisome protein FtsK as a potential direct interaction partner of DmxA. However, while FtsZ localizes at mid-cell in ~50% of cells[53] and FtsK in ~15% of cells[54], DmxA only localizes to mid-cell in ~5% of cells, indicating that FtsK may not directly recruit DmxA. Interestingly, FtsK is important for recruiting the late-division proteins FtsQ, -L and -B[60], which are bitopic membrane proteins, and DmxA is encoded in an operon with FtsB (Fig. S3d, e). Therefore, it remains a possibility that DmxA could be recruited to the division site by these protein(s). Of note, FtsQ, -L, and -B have short cytoplasmic tails[60], making them difficult targets for proximity labeling.

Several lines of evidence support that DmxA is activated upon recruitment to the division site. First, the timing of DmxA localization to the division site and the burst in c-di-GMP perfectly correlate. Second, DmxA is required and sufficient for the burst in c-di-GMP. Third, DmxA^ΔTMD does not localize to the division site and is not active in vivo; however, the protein has DGC activity in vitro. Although we cannot rule out that DmxA^ΔTMD is less active than full-length DmxA, these observations jointly also support that DmxA is activated at the division site. Finally, the observations that the steady-state level of DmxA-mVenus is constant over the cell cycle and DmxA accumulates at the same level in cephalexin-treated cells and FtsZ-depleted cells as in untreated cells, strongly indicate that DmxA accumulation is not cell cycle-regulated and that DmxA activity is not regulated by the total cellular concentration. Based on these observations, we suggest that DmxA DGC activity, upon recruitment to the division site, is switched on either by interacting with protein(s) of the divisome or, alternatively, the high local DmxA concentration stimulates the formation of the enzymatically active dimer. We speculate that the low-affinity I-site allows DmxA to synthesize high concentrations of c-di-GMP, and may solely be relevant at very high concentrations to avoid excessive over-production of c-di-GMP. Upon completion of cytokinesis, the DmxA cluster disintegrates, likely subsequent to the disassembly of the divisome. As a consequence, c-di-GMP synthesis ceases, and its level decreases rapidly. Interestingly, none of the six predicted PDEs of *M. xanthus* have been implicated in motility[39]. In the future, it will be interesting to determine which PDE(s) are involved in the rapid decrease in the c-di-GMP concentration.

Despite only being active during a brief period of the cell cycle, DmxA is essential for WT motility behavior. During this brief period, DmxA guarantees the symmetric incorporation and allocation at the

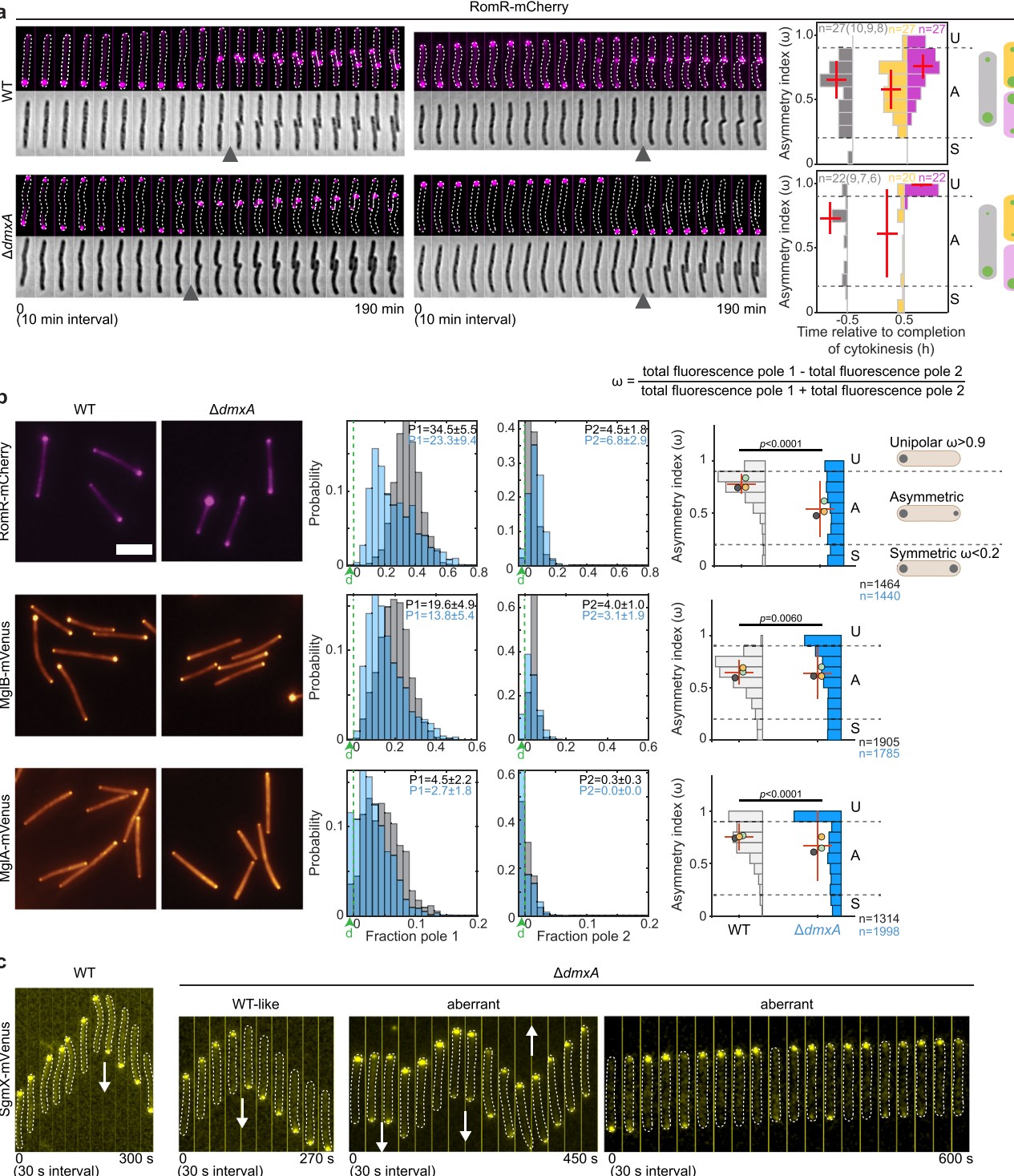

$$\omega = \frac{\text{total fluorescence pole 1 - total fluorescence pole 2}}{\text{total fluorescence pole 1 + total fluorescence pole 2}}$$

**Fig. 6 | DmxA is essential for the symmetric allocation of polarity proteins to the daughters during cytokinesis. a** RomR-mCherry localization during the cell cycle. Left panels, epifluorescence and phase-contrast images from representative time-lapse microscopy of WT and Δ*dmxA* cells. Images were recorded every 10 min; arrowheads indicate completion of cytokinesis. Right panels, histogram quantification of RomR-mCherry localization pattern based on time-lapse microscopy. RomR-mCherry polar fluorescence was quantified 30 min before and after division in the predivisional cell (gray) and daughters (yellow/pink) and ω was calculated as indicated. Daughter cells were sorted based on their ω with the most symmetric in yellow and the other in pink. Localization patterns were binned as unipolar, asymmetric, and symmetric from the ω values as in Fig. 5b. Line and error bars

indicate the median±MAD of the indicated number of cells from three biological replicates, number of cell division events per replicate in brackets. Note that in the Δ*dmxA* mutant, two daughters only had a diffused signal. Schematics show dominant localization patterns of RomR-mCherry in WT and Δ*dmxA* cells during the cell cycle. **b** Localization of RomR-mCherry, MglB-mVenus, and MglA-mVenus in WT and Δ*dmxA* cells. Left panels, representative snapshot images. Analysis of snapshot images of WT (gray) and Δ*dmxA* (blue) were done as in Fig. 5b. **c** SgmX-mVenus localization in representative moving WT and Δ*dmxA* cells. Images were recorded every 30 s. White arrows indicate reversals. Source data are provided as a Source Data file.

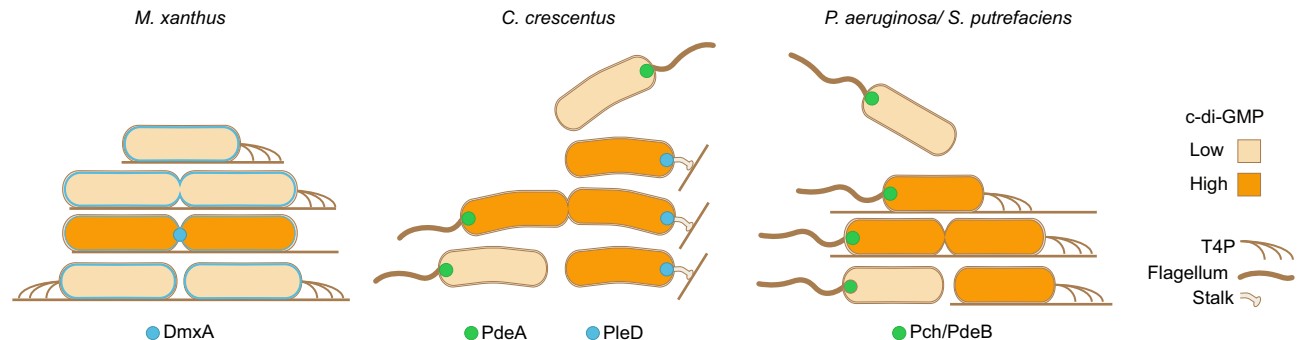

**Fig. 7 | Deployment of DGC and/or PDEs to distinct subcellular locations establishes deterministic programs hardwired into the cell cycle to generate or minimize phenotypic heterogeneity.** In *M. xanthus*, DmxA (blue) localizes to the membrane. During division, it localizes to and is switched on at the division site creating a c-di-GMP burst that ensures similar daughter cells. In *C. crescentus*, the flagellated, swarmer cell has low c-di-GMP due to the activity of the PDE PdeA at the flagellated pole (green). Upon differentiation to the surface-associated stalked cell, the c-di-GMP level increases due to the activity of the DGC PleD at the stalked pole (blue). In predivisional cells, PdeA and PleD localize to opposite poles, giving rise to a swarmer cell with low c-di-GMP and a stalked cell with high c-di-GMP upon division. In *P. aeruginosa*/*S. putrefaciens*, the flagellated, swimming cell, has low c-di-GMP due to the activity of the PDE Pch/PdeB at the flagellated pole (green). Upon surface contact, c-di-GMP increases, but the involved DGC(s) remain to be identified. High c-di-GMP stimulates T4P formation and surface adhesion. During division, the flagellated pole inherits the PDE, creating a flagellated, swimming daughter with low c-di-GMP and a surface-adhered, piliated daughter with high c-di-GMP.

nascent and new poles of PilQ of the T4PM and RomR of the polarity module. Because PilQ and RomR are at the base of the assembly of the core T4PM and the polar localization of the proteins of the polarity module, respectively, we suggest that the defects in PilQ and RomR polar localization during cytokinesis in the absence of DmxA cause the observed misincorporation of the core T4PM and the remaining polarity regulators, respectively. Because the polarity defects that arise during one division in cells lacking DmxA are not fully corrected until the next division, and mutants with aberrant localization of the polarity proteins have aberrant cell behaviors with altered reversal frequencies[24,28,29,31,32,35,38], we suggest that the aberrant T4P-dependent motility behavior in the absence of DmxA is the result of dual defects, i.e., the defects in the polar incorporation of the core T4PM and in the localization of polarity proteins. We also speculate, but have not shown, that the aberrant gliding behavior is not only a consequence of the mislocalized polarity proteins but also involves defects in the polar incorporation of structural proteins of the Agl/Glt machine.

How, then, does the burst in c-di-GMP ensure the correct incorporation and allocation of polarly localized motility proteins and regulators? The effects of changing c-di-GMP levels are implemented by the binding of c-di-GMP to downstream effectors[2,3]. Because polar PilQ incorporation depends on its peptidoglycan-binding AMIN domains[61], we suggest that the c-di-GMP burst brings about the localization of a landmark protein, which possibly binds peptidoglycan, at the nascent and new poles that assist polar recruitment of PilQ. The cytoplasmic RomR protein, by an unknown mechanism, localizes polarly in the absence of the other polarity proteins[51]. The symmetric allocation of RomR to the daughters involves a polarity reset involving three steps, i.e., RomR release from the old poles, its recruitment to the division site, and its symmetric allocation to the two daughters. In the absence of DmxA, RomR was still released from the old poles in most cells, suggesting that this step is independent of DmxA and c-di-GMP. However, RomR was either not recruited to the division site or, if it was recruited, then it was not evenly allocated to the two daughters. Therefore, we suggest that c-di-GMP also brings about the localization of a landmark protein at the division site that is recognized by RomR. Because the defects in PilQ and RomR polar localization are not fully corrected between division events, we also suggest that these landmark(s) may only be transiently active. This mechanism is supported by the observations that c-di-GMP regulates the transient polar localization of effectors with landmark function, including TipF in *C.*

*crescentus* that marks the incorporation site of the flagellum at the new pole[62,63], and FimX in *Pseudomonas aeruginosa* that recruits PilB to promote T4P-dependent motility[6]. In future experiments, it will be important to identify the effector(s) involved in the response to DmxA-generated c-di-GMP and to address whether these effector(s) serve as landmark(s) or to recruit landmark(s). Nonetheless, we speculate that an advantage of engaging a DGC in setting up correct cell polarity during cytokinesis could be that c-di-GMP would allow the transient function of effector(s)/polar landmark(s). Finally, signaling by c-di-GMP has been proposed to involve either a global c-di-GMP pool or a local c-di-GMP pool[64,65]. Because the c-di-GMP pool generated by DmxA is sensed by the cdGreen2 biosensor, we infer that DmxA contributes to a global pool of c-di-GMP.

Several bacteria that alternate between a planktonic, flagellum-dependent swimming lifestyle and a surface-associated lifestyle, harness c-di-GMP to deterministically generate phenotypically distinct daughters during division[4,11]. In *C. crescentus*, *P. aeruginosa*, and *Shewanella putrefaciens*, the programs driving the generation of this heterogeneity rely on the asymmetric deployment of c-di-GMP metabolizing enzyme(s) to the daughters during cell division, i.e., either the relevant DGC and PDE localize to opposite cell poles or a PDE localizes unipolarly[7,45,55,66–72] (Fig. 7). Consequently, one daughter has low c-di-GMP and becomes the flagellated swimming daughter, while the other daughter has high c-di-GMP and becomes the surface-associated daughter. By contrast, *M. xanthus* places the DGC DmxA at the division site, thereby enabling the formation of mirror-symmetric and phenotypically similar daughters. Thus, by deploying c-di-GMP synthesizing and degrading enzymes to distinct subcellular locations, bacteria harness c-di-GMP to establish deterministic programs that are hardwired into the cell cycle to generate or, as shown here, minimize phenotypic heterogeneity (Fig. 7).

Similar to stochastically generated phenotypic heterogeneity[73–75], the deterministic generation of phenotypic heterogeneity has been suggested to be part of a bet-hedging strategy in which the diversification of phenotypes optimizes the survival of the population and/or a division of labor strategy by enabling the colonization of multiple habitats in parallel[7,69,72,76]. Because *M. xanthus* translocates on surfaces in large cooperative swarms in which the motility of individual cells is highly coordinated, we speculate that reducing phenotypic heterogeneity during cell division optimizes its ability to perform its motility-dependent social behaviors.

## Methods

### Bacterial strains and growth media

*M. xanthus* cells were grown at 32 °C in 1% CTT (1% (w/v) Bacto Casitone (Gibco) in TPM buffer (10 mM Tris-HCl pH 8.0, 1 mM $K_2HPO_4$/$KH_2PO_4$ pH 7.6, and 8 mM $MgSO_4$)) liquid medium or on 1.5% agar supplemented with 1% CTT[77]. Oxytetracyline and kanamycin at concentrations of 10 and 50 µg ml⁻¹, respectively were added when needed. Cephalexin was added to a concentration of 35 µg ml⁻¹ in a liquid medium and 20 µg ml⁻¹ on agarose. All *M. xanthus* strains are derivatives of the WT strain DK1622[18]. *M. xanthus* strains, plasmids, and oligonucleotides used in this work are listed in Tables S3–S5, respectively. In-frame deletions or gene replacements were generated as described[78], plasmids were integrated in a single copy by site-specific recombination at the Mx8 *attB* site or by homologous recombination at the *MXAN_18-19* site or at the endogenous locus. All in-frame deletions and plasmid integrations were verified by PCR. Plasmids were propagated in *Escherichia coli* Mach1, which was grown at 37 °C in lysogeny broth (LB) medium (10 mg tryptone ml⁻¹, 5 mg yeast extract ml⁻¹, and 10 mg NaCl ml⁻¹) supplemented when required with kanamycin (50 µg ml⁻¹), tetracycline (25 µg ml⁻¹), or ampicillin (100 µg ml⁻¹).

### Motility assays

Population-based motility assays were performed as described[79]. Briefly, exponentially growing suspension cultures were harvested (3 min, 8000 × *g*, room temperature (RT)) and resuspended in 1% CTT to a calculated density of $7 \times 10^9$ cells ml⁻¹. About 5 µl aliquots of cell suspensions were spotted on 0.5% agar (Invitrogen) and 1.5% agar (Gibco) supplemented with 0.5% CTT and incubated at 32 °C. Cells were imaged at 24 h using a M205FA Stereomicroscope (Leica) and a DMi8 inverted microscope (Leica) equipped with a Hamamatsu ORCA-Flash4.0 V2 digital CMOS C11440 camera (Hamamatsu Photonics) and DFC9000 GT camera (Leica), respectively. To visualize single cells moving by T4P-dependent motility, 5 µl exponentially growing cells in suspension were placed in a 24-well polystyrene plate (Falcon). After 10 min incubation in the dark at RT, 200 µl of 1% methylcellulose in MMC buffer (10 mM MOPS, 4 mM $MgSO_4$, 2 mM $CaCl_2$, pH 7.6) were added, and cells incubated for 30 min in the dark at RT. Cells were imaged for 15 min with 30 s intervals. To visualize individual cells moving by gliding, exponentially growing cells in suspension were diluted to $3 \times 10^8$ and 5 µl spotted on 1.5% agar (Gibco) supplemented with 0.5% CTT and immediately covered with a coverslide. Cells were incubated 2 h at 32 °C and then visualized for 15 min with 30 s intervals at RT. Cells were imaged using a DMi8 Inverted microscope and a DFC9000 GT camera. Images were analysed using ImageJ[80].

### Analysis of EPS accumulation

Colony-based EPS assays were performed as described previously[40,81]. Briefly, exponentially growing cells were harvested (3 min, 6000 × *g*, RT) and resuspended in 1% CTT to a calculated density of $7 \times 10^9$ cells ml⁻¹. About 20-µl aliquots were placed on 0.5% agar plates supplemented with 0.5% CTT and 10 or 20 µg ml⁻¹ of Trypan blue or Congo red, respectively. Plates were incubated at 32 °C and imaged at 24 h.

### Negative stain transmission electron microscopy

About 10 µl of *M. xanthus* cells exponentially grown in suspension were placed on one side of the electron microscopy grid (Plano) and incubated at RT for 40 min. To avoid evaporation during this step, the grid was incubated in humid air. The liquid was blotted through the grid by capillarity by applying the side of the grid on Whatman paper. Cells were washed three times with 10 µl of double-distilled water and stained with UA-Zero EM Stain (Plano) (diluted to 0.25% (v/v) in double-distilled water). After 1 min incubation, the liquid was removed and cells were washed once with double-distilled water to remove excess staining solution. Transmission electron microscopy was done with a JEOL JEM-1400 electron microscope at 100 kV.

### Immunoblot analysis

Immunoblots were performed as described[82]. Rabbit polyclonal α-LonD (dilution: 1:5000)[83], α-PilA (dilution: 1:2000)[83], α-PilC (dilution: 1:2000)[15], α-FtsZ (dilution: 1:25,000)[53], α-mCherry (dilution: 1:1000) (BioVision), α-FLAG (dilution, 1:2000) (Rockland) and α-RFP (dilution 1:2000) (Rockland), were used together with horseradish peroxidase-conjugated goat α-rabbit immunoglobulin G (dilution: 1:15,000) (Sigma) as secondary antibody. Mouse α-GFP antibodies (dilution: 1:2000) (Roche) were used together with horseradish peroxidase-conjugated sheep α-mouse immunoglobulin G (dilution: 1:2000) (GE Healthcare) as a secondary antibody. Streptavidin-HRP (dilution, 1:4000) (IBA) was prepared in 3% BSA as described[56]. Blots were developed using Luminata Forte Western HRP Substrate (Millipore) on a LAS-4000 imager (Fujifilm).

### T4P shear-off assay

T4P were sheared from *M. xanthus* cells following a slightly modified protocol[83]. Briefly, cells grown for three days on 1% CTT 1.5% agar plates at 32 °C were scraped off the agar surface and resuspended in pili resuspension buffer (100 mM Tris-HCl pH 7.6, 150 mM NaCl) (1 ml per 60 mg cells). Cell suspensions were vortexed for 10 min at maximum speed. For determining the cellular PilA level, a 100 µl aliquot was harvested (10 min, 21,100 × *g*, 4 °C) and the pellet resuspended in 200 µl 1 × sodium dodecyl sulfate (SDS) buffer (60 mM Tris-HCl pH 6.8, 2% (w/v) SDS, 10% (v/v) glycerol, 0.1 M dithiothreitol, 5 mM EDTA, 0.005% Bromophenol Blue), and denatured at 95 °C for 10 min. The remaining cell suspension was centrifuged three times (20 min, 21,000 × *g*, 4 °C) to remove cell debris. Pili in the cell-free supernatant were precipitated by adding 10 × pili precipitation buffer (final concentrations: 100 mM $MgCl_2$, 2% (w/v) PEG 6000, 100 mM Tris-HCl pH 7.6, 150 mM NaCl) for at least 2 h at 4 °C and then harvested (21,000 × *g*, 30 min, 4 °C). The pellet was resuspended in 1× SDS buffer (2 µl per mg vortexed cells) and boiled for 10 min at 95 °C. Samples were separated by SDS-PAGE and analyzed for PilA accumulation by immunoblot using PilA antibodies. Blots were developed as described above.

### Operon mapping

Mapping of the *ftsB-dmxA* operon was performed as described[61]. Briefly, $1 \times 10^9$ WT cells from an exponentially growing suspension culture were harvested (3 min, 8000 × *g*, RT) and resuspended in 200 µl lysis buffer (100 mM Tris-HCl pH 7.6, 1 mg ml⁻¹ lysozyme). After incubation at 25 °C for 5 min, cells were lysed and RNA purified using the Monarch Total RNA Miniprep Kit (NEB) according to the manufacturer's instructions, except that the on-column DNase treatment was omitted. RNA was eluted in RNase-free water, treated with Turbo DNase (Invitrogen), purified using the Monarch RNA Cleanup Kit (50 µg) (NEB), and eluted in RNase-free water. About 1 µg of RNA was used for cDNA synthesis using the LunaScript RT SuperMix Kit (NEB) with and without reverse-transcriptase. cDNA, the mock reaction without reverse-transcriptase, or genomic DNA were used as templates for PCR using the primers listed in Table S5.

### Cell length determination

About 5-µl aliquots of exponentially growing suspension cultures were spotted on 1% agarose supplemented with 0.2% CTT. Cells were immediately covered with a coverslide, and imaged using a DMi8 Inverted microscope and DFC9000 GT camera. To assess cell length, cells were segmented using Omnipose[84], segmentation was manually curated using Oufti[85], analysed using Matlab R2020a (The MathWorks) and plotted using GraphPad Prism (GraphPad Software, LLC).

### Fluorescence microscopy

In all time-lapse microscopy experiments except for those involving SgmX-mVenus, cells were visualized as in ref. 86 with slight

modifications. Briefly, 5 µl exponentially growing cells in suspension were placed on a glass coverslide attached to a plastic frame. Cells were covered with a thick 1% agarose pad supplemented with 0.2% CTT, the pad was sealed with parafilm to reduce evaporation, and cells were imaged after 180 min. To avoid that cells would move out of the field of view, all strains contained the Δ*gltB* mutation. To clearly distinguish leading and lagging cell poles of cells moving by T4P-dependent motility, time-lapse microscopy experiments involving SgmX-mVenus, were done on Chitosan-coated µ-Dishes (Ibidi) as described[87]. Briefly, a 100 µl-aliquot of exponentially growing cells was diluted in 900 µl MC7 buffer (10 mM MOPS pH 7.0, 1 mM CaCl₂), spotted on the chitosan-coated µ-Dish, and imaged after 30 min. Snapshot microscopy images were captured from cells on a 1% agarose pad supplemented with 0.2% CTT (biosensor and DmxA-mVenus) or on Chitosan-coated µ-Dishes (all other fluorescent proteins). Cells were imaged using a DMi8 inverted microscope and a Hamamatsu ORCA-Flash4.0 V2 Digital CMOS C11440 or a DFC9000 GT camera. Data were analysed using Oufti[85], Metamorph® v 7.5 (Molecular Devices), Matlab, and ImageJ[80]. DmxA-mVenus clusters and constrictions were identified manually. DmxA-mVenus total cellular fluorescence in snapshots and time-lapse recordings was analysed by segmenting the cells using Oufti[85], and the background corrected normalized cellular fluorescence was calculated using Matlab.

To identify and analyse polar clusters in snapshots, we used a custom-made Matlab script[31]. Briefly, cells were segmented, and polar clusters were identified as having an average fluorescence signal of 1.5 SD (MglA) or 2 SD (all other proteins), above the mean cytoplasmic fluorescence and a size of three or more pixels. For each cell with polar clusters, an asymmetry index (ω) was calculated as:

$$\omega = \frac{\text{total fluorescence at pole 1} - \text{total fluorescence at pole 2}}{\text{total fluorescence at pole 1} + \text{total fluorescence at pole 2}} \quad (1)$$

Pole 1 was assigned to the pole with the highest fluorescence. The localization patterns were binned from the ω values as follows: unipolar (ω > 0.9), bipolar asymmetric (0.9 ≥ ω ≥ 0.2), and bipolar symmetric (0.2 > ω). Diffuse localization was determined when no polar signal was detected. The polar fluorescence of SgmX-mVenus in moving cells was followed manually.

For the analysis of single-cell cdGreen2 fluorescence, cells were segmented using Omnipose[84], and the segmentation was manually curated using Oufti[85]. For normalization, the average cellular fluorescence of each cell in the green channel (cdGreen2) was divided by the red channel (mScarlet-I) using Matlab.

## C-di-GMP quantification

C-di-GMP levels were determined as described[88]. Briefly, 4 ml of exponentially growing cells were harvested by centrifugation (20 min, 2500 × g, 4 °C). Cells were mixed with 300 µl ice-cold extraction buffer (high-pressure liquid chromatography [HPLC]-grade acetonitrile-methanol-water [2:2:1, v-v:v]), and incubated 15 min at 4 °C to quench metabolism. Extraction was performed at 95 °C for 10 min, samples were centrifuged (10 min, 21,130 × g, 4 °C), and the supernatant containing extracted metabolites was transferred to a new Eppendorf tube. The pellet was washed with 200 µl extraction buffer and centrifuged (10 min, 21,130 × g, 4 °C). This step was repeated. The residual pellet containing proteins was kept, and the three supernatants from the extraction and the two washing steps containing c-di-GMP were pooled and evaporated to dryness in a vacuum centrifuge. Subsequently, the samples with c-di-GMP were dissolved in HPLC-grade water for analysis by liquid chromatography-coupled tandem mass spectrometry (LC-MS/MS). In parallel, to determine the protein concentration for each sample, the residual pellets were resuspended in 800 µl 0.1 M NaOH, and heated for 15 min at 95 °C until dissolution.

Protein levels were determined using a 660 nm Protein Assay (Pierce) following the manufacturer's instructions.

## Protein purification

For expression and purification of MalE-tagged DmxA variants, proteins were expressed in *E. coli* Rosetta DE3 growing in 5052-Terrific-Broth[89] (0.5% (v/v) glycerol, 0.05% (w/v) glucose, 0.2% (w/v) lactose, 2.4% (w/v) yeast extract, 2% (w/v) tryptone, 25 mM Na₂HPO₄, 25 mM KH₂PO₄, 50 mM NH₄Cl, 5 mM Na₂SO₄, and 2 mM MgSO₄) auto-induction medium supplemented with 25 µg ml⁻¹ chloramphenicol and 100 µg ml⁻¹ carbenicillin. Cells were grown at 37 °C until OD₆₀₀ = 1, shifted to 18 °C and further incubated overnight. Cells were harvested and resuspended in MalE-lysis buffer (100 mM Tris-HCl pH 7.2, 500 mM NaCl, 10 mM MgCl₂, 5 mM DTT) supplemented with EDTA-free protease inhibitor cocktail (Roche) and lysed by sonication for ten cycles of 30 pulses of sonication and 30 s breaks using a Hielscher UP200st set to pulse = 70%, amplitude = 70%. The lysate was cleared by centrifugation (16,000 × g, 4 °C, 30 min) and loaded onto a 5 ml HighTrap MBP column (Cytiva) using an Äkta-Pure system (GE Healthcare). The column was washed with 10 column volumes of lysis buffer and protein eluted with MalE-elution buffer (100 mM Tris-HCl pH 7.2, 500 mM NaCl, 10 mM MgCl₂, 5 mM DTT, and 10 mM Maltose). The elution fractions containing MalE-DmxA variants were pooled and loaded on a HiLoad 16/600 Superdex 200 pg (GE Healthcare) SEC column, which was pre-equilibrated with SEC buffer (50 mM Tris-HCl pH 7.2, 250 mM NaCl, 10 mM MgCl₂, 5 mM DTT, 5% glycerol (v/v)) and protein was eluted using SEC buffer. Subsequently, protein was either used fresh or snap-frozen in SEC buffer. The SEC column was calibrated using Ribonuclease A, Carbonic Anhydrase, Conalbumin (Gel Filtration Calibration Kit LMW, Cytiva) and Ferritin (Gel Filtration Calibration Kit HMW, Cytiva).

## DGC activity assay

DGC activity assays were performed, using the EnzCheck®Pyrophosphate Assay Kit (Thermo) as described[90]. Briefly, the release of inorganic pyrophosphate during c-di-GMP synthesis was followed by measuring the absorbance change at 360 nm in a Tecan M200 pro, in 30 s intervals for 1 h. Reactions contained 1 µM protein, and 50 µM GTP. Inhibition reactions were fit in GraphPad Prism to the equation

$$V_{[cdG]} = V_0 / (1/1 + ([cdG]/K_i)^h) \quad (2)$$

where $V_0$ represents the reaction velocity in the absence of c-di-GMP, [cdG] the concentration of c-di-GMP in the reaction, $K_i$ the inhibitory constant, and h the Hill coefficient.

## In vitro nucleotide binding assay

C-di-GMP binding was determined by Bio-Layer Interferometry using the BLItz system (ForteBio)[91] and a Streptavidin SA biosensor (ForteBio). Briefly, 500 nM biotinylated c-di-GMP (Biolog) in SEC buffer supplemented with 0.1% (v/v) Tween-20 was immobilized onto the biosensors for 120 s, and unbound molecules washed off for 30 s. Association and dissociation of a protein were carried out for 120 and 120 s, respectively. The binding was fitted to the "One site – Total" binding model in GraphPad Prism.

## Proximity labeling

Proximity labeling, including shotgun proteomics analysis was done as described[56]. Briefly, 50 ml of exponentially growing cell suspension were incubated with 100 µM biotin and 35 µg ml⁻¹ cephalexin. After 4 h, cells were harvested by centrifugation (8000 × g, 10 min, 4 °C), resuspended in 600 µl RIPA buffer (50 mM Tris-HCl pH 7.0, 150 mM NaCl, 0.5% (w/v) sodium deoxycholate, 0.2% (w/v) SDS, and 1% (v/v) Triton-X100) supplemented with protease inhibitor cocktail (Roche) and lysed by 30 pulses of sonication using a Hielscher UP200st set to pulse

50% and amplitude 50%. SpinTrap G-25 columns (Cytiva) were used to remove an excess of biotin from the cleared lysate. To enrich biotinylated proteins, 500 µl of each sample was incubated for 1 h at 4 °C with 50 µl streptavidin magnetic beads (Pierce). The beads were washed three times with 1 ml RIPA buffer, twice with 1 ml 1 M KCl, and three times with 1 ml 50 mM Tris-HCl pH 7.6. Finally, proteins were eluted using on-bead digest as described[83]. Briefly, 100 µl elution buffer 1 (100 mM ammonium bicarbonate, 1 µg trypsin (Promega)) was added to each sample. After 30 min incubation at 30 °C, the supernatant containing the digested proteins was collected. Beads were washed twice with elution buffer 2 (10 mM ammonium bicarbonate, 5 mM Tris(2-carboxyethyl)phosphine hydrochloride (TCEP)) and added to the first elution fraction. Digestion continued overnight at 30 °C. Next, the peptides were incubated with 10 mM iodoacetamide for 30 min at 25 °C in the dark. Prior to LC-MS analysis, peptide samples were desalted using C18 solid phase extraction spin columns (Macherey-Nagel). Peptide mixtures were then analysed using LC-MS on an Exploris 480 instrument connected to an Ultimate 3000 RSLCnano and a nanospray flex ion source (all Thermo Scientific). A detailed description of the LC-MS parameters are described in ref. 56. The following separating gradient was used: 98% solvent A (0.15% formic acid) and 2% solvent B (99.85% acetonitrile, 0.15% formic acid) to 30% solvent B over 40 min at a flow rate of 300 nl/min. Peptides were ionized at a spray voltage of 2.3 kV, and ion transfer tube temperature set at 275 °C, 445.12003 m/z was used as internal calibrant. The data acquisition mode was set to obtain one high-resolution MS scan at a resolution of 60,000 full width at half maximum (at m/z 200) followed by MS/MS scans of the most intense ions within 1 s (cycle 1 s). The ion accumulation time was set to 50 ms (MS) and 50 ms at 17,500 resolution (MS/MS). The automatic gain control (AGC) was set to $3 \times 10^6$ for MS survey scan and $2 \times 10^5$ for MS/MS scans. MS raw data were then analysed with MaxQuant[92], and an *M. xanthus* UniProt database[93]. MaxQuant was executed in standard settings without "match between runs" option. The search criteria were set as follows: full tryptic specificity was required (cleavage after lysine or arginine residues); two missed cleavages were allowed; carbamidomethylation (C) was set as fixed modification; oxidation (M) and deamidation (N,Q) as variable modifications. The MaxQuant proteinGroups.txt file was further processed by the SafeQuant R package for statistical analysis[94].

## Total proteome analysis

The total proteome of *M. xanthus* cells grown in suspension culture was determined following a slightly modified protocol of ref. 83. Briefly, 2 ml of exponentially growing suspension cultures were harvested (8000 × *g*, 3 min, RT). Cells were resuspended in 1 ml PBS and harvested again. Subsequently, the supernatant was discarded and the pellet snap-frozen in liquid nitrogen. The pellet was suspended in 150 µl 2% sodium lauryl sulfate (SLS) and proteins precipitated using acetone. For digestion, samples were resuspended in 0.5% SLS with 1 µg trypsin (Promega) and incubated for 30 min at 30 °C, subsequently 5 mM TCEP were added to the suspension and further incubated overnight. Following, acetylation using 10 mM iodoacetamide for 30 min at 25 °C in the dark, the peptides were desalted using C18 solid phase extraction. For label-free protein quantification, peptide mixtures were analysed using LC-MS. The data were acquired in data-independent acquisition mode and the MS raw data analysed by DIA-NN as described[95,96]. Data were further analysed and plotted using Python (3.7). The mass spectrometry proteomics data of whole cell proteomics and proximity labeling experiments have been deposited to the ProteomeXchange Consortium[97] via the PRIDE[98] partner repository with the dataset identifier PXD049046.

## Bioinformatics

The KEGG database[99] was used to assign functions to proteins, identify orthologs of *M. xanthus* proteins using a reciprocal best BlastP hit method and collect the 16s ribosomal RNA sequence of fully sequenced myxobacteria (Table S6). Protein domains were identified using InterPro[100], SMART[101], and the predicted AlphaFold structures. The DmxA protein sequence without the N-terminal transmembrane helices (amino acid 1–209) was used for AlphaFold-Multimer modeling via ColabFold (1.5.0)[102–104]. The predicted local distance difference test (pLDDT) and predicted alignment error (pAE) graphs of the five models generated were made using a custom Matlab script. Models were ranked based on combined pLDDT and pAE values, with the best-ranked models used for further analysis and presentation. Per the residue model, accuracy was estimated based on pLDDT values (>90, high accuracy; 70–90, generally good accuracy; 50–70, low accuracy; <50, should not be interpreted)[102]. Relative domain positions were validated by pAE. The pAE graphs indicate the expected position error at residue X if the predicted and true structures were aligned on residue Y; the lower the pAE value, the higher the accuracy of the relative position of residue pairs and, consequently, the relative position of domains/subunits/proteins[102]. PyMOL version 2.4.1 (http://www.pymol.org/pymol) was used to analyse and visualize the models. The phylogenetic tree was prepared using the 16s ribosomal RNA sequence of fully sequenced myxobacteria in MEGA[105] using the Neighbor-Joining method[106]. Bootstrap values (500 replicates) are shown next to the branches[107]. RNA-seq. data were plotted using the BioMap function in Matlab. The base-by-base alignment coverage of RNA-seq and Cappable-seq reads of[108] were plotted for each position.

## Statistics

Colony expansion, c-di-GMP measurements, and immunoblot quantifications were analysed using the two-sided Student's *t*-test in GraphPad Prism. Single-cell speed and cell length distributions were analysed using the two-sided Mann–Whitney test in GraphPad Prism. Single-cell reversal assays were analysed using the One-Way ANOVA function with Fishers LSD post hoc test in GraphPad Prism. Asymmetry indexes (ω) were analysed using the rank-sum function (two-sided Mann–Whitney test) in Matlab. MAD was used as a measure of data variability and calculated based on the formula

$$MAD = median(|x_i - \tilde{x}|). \tag{3}$$

In the analysis of enriched proteins in proximity labeling experiments, *p*-values were calculated using eBayes moderated *t*-statistics[109].

## Plasmid construction

All oligonucleotides used are listed in Table S5. All constructed plasmids were verified by DNA sequencing.

pMP072 (for in-frame deletion of *dmxA*): Up- and downstream fragments were amplified from genomic DNA using the primer pairs 3705_A/3705_B and 3705_C/3705_D, respectively. Subsequently, the up- and downstream fragments were used as a template for an overlapping PCR with the primer pair 3705_A/3705_D to generate the AD fragment. The AD fragment was digested with XbaI and KpnI, cloned in pBJ114, and sequenced. Of note, in order to successfully delete *dmxA*, the length of the flanking regions was increased compared to the plasmid (pTP126) used in ref. 39.

pMP092 (plasmid for expression of *dmxA-mVenus* under control of P_nat from the *attB* site): The P_nat *dmxA* fragment was amplified from gDNA using the primer pair 3704 prmt forw +XbaI/3705_rev no stop 1. The mVenus fragment was amplified using pLC20[31] as DNA template and the primer pair 3705_mVenus fw/mVenus_Kpn rev. To generate the full-length insert, an overlapping PCR using the two fragments as DNA templates and the primer pair 3704 prmt forw +XbaI/mVenus_Kpn rev was performed. The fragment was digested with KpnI and XbaI, cloned into pSWU30, and sequenced.

pMP093 (plasmid for replacement of *dmxA* with *dmxA-mVenus* in the native site): Up- and downstream fragments were amplified using

genomic DNA from *M. xanthus* DK1622 as DNA template and the primer pairs 3705_native forw/3705_rev no stop 1 and 3705_native middle fw/3705_native rev, respectively. The mVenus fragment was amplified using pMP092 as DNA template and the primer pair 3705_mVenus fw/3705_native middle rev. To generate the full-length insert, an overlapping PCR using the three fragments as DNA templates and the primer pair 3705_native forw/3705_native rev was performed. The fragment was digested with KpnI and XbaI, cloned into pBJ114, and sequenced.

pMP164 (plasmid for expression of *dmxA-mVenus* from the native site in a Δ*dmxA* strain): Up- and downstream fragments were amplified using genomic DNA from *M. xanthus* DK1622 as a template and the primers 3705_A/3705_rev no stop 1 and 3705_native middle fw/3705_D respectively. mVenus was amplified from pMP093 as a template and the primers 3705_mVenus fw/3705_native middle rev. To generate the AD insert, an overlapping PCR using the three fragments as a DNA template and the primer pair 3705_A/3705_D was performed. The AD fragment was digested with KpnI and XbaI, cloned into pBJ114, and sequenced.

pMP165 (plasmid for expression of *dmxA*$^{E626A}$-*mVenus* from the native site in a Δ*dmxA* strain): The mutation E626A was introduced into the plasmid pMP164 by using the primer pairs 3705_A/3705 E626A (−) and 3705 E626A (+)/3705_D and pMP164 as a DNA template for the PCR. To generate the AD insert, an overlapping PCR using both fragments as a DNA template and the primer pair 3705_A/3705_D was performed. The AD fragment was digested with KpnI and XbaI, cloned into pBJ114, and sequenced.

pMP095 (plasmid for replacement of *dmxA* with *dmxA*$^{R615A}$-*mVenus* in the native site): Up- and downstream fragments were amplified using pMP093 as DNA template and the primer pairs 3705_native forw/3705 R615A (−) and 3705 R615A (+)/3705_native rev, respectively. To generate the full-length insert, an overlapping PCR using the two fragments as DNA templates and the primer pair 3705_native forw/3705_native rev was performed. The fragment was digested with KpnI and XbaI, cloned into pBJ114, and sequenced.

pKH02 (plasmid for deletion of *mVenus* in the *dmxA*$^{R615A}$-*mVenus* strain): Up- and downstream fragments were amplified using pMP095 as DNA template and the primer pairs 3705_native forw/3705_del_mVenus_(−) and 3705_del_mVenus_(+)/3705_native rev, respectively. To generate the full-length insert, an overlapping PCR using the two fragments as DNA templates and the primer pair 3705_native forw/3705_native rev was performed. The fragment was digested with KpnI and XbaI, cloned into pBJ114, and sequenced.

pMP175 (for generation of an in-frame deletion of the GAF domains of native *dmxA*): Up- and downstream fragments were amplified from pMP164 using the primer pairs 3705_PpilA forw/3705_B (GAF × 1) and 3705_C (GAF × 2)/3705_D (GAF × 2), respectively. Subsequently, the up- and downstream fragments were used as a template for an overlapping PCR with the primer pair 3705_PpilA forw/3705_D (GAF × 2) to generate the AD fragment. The AD fragment was digested with XbaI and KpnI, cloned in pBJ114, and sequenced.

pMP179 (for generation of an in-frame deletion of the TMD domains of native *dmxA*): Up- and downstream fragments were amplified from pMP164 using the primer pairs 3705_A/3705_B3 (TMD) and 3705_C3 (TMD)/3705_D (TMD)2, respectively. Subsequently, the up- and downstream fragments were used as a template for an overlapping PCR with the primer pair 3705_A/3705_D (TMD)2 to generate the AD fragment. The AD fragment was digested with KpnI and XbaI, cloned in pBJ114, and sequenced.

pMP182 (plasmid for expression of *dmxA*$^{TMD (1-167)}$-*mVenus* from the native site in a Δ*dmxA* strain): Up- and downstream fragments were amplified using pMP164 as a DNA template and the primers 3705_A/3705_TMH rev and 3705_mVenus fw2/3705_D respectively. To generate the AD insert, an overlapping PCR using the two fragments as a DNA template and the primer pair 3705_A/3705_D was performed. The AD

fragment was digested with KpnI and XbaI, cloned into pBJ114, and sequenced.

pMH113 (plasmid for expression of *malE-dmxA*$^{WT (223-722)}$): The *dmxA*$^{223-722}$ insert was amplified using genomic DNA from *M. xanthus* DK1622 as DNA template and the primer pair MalE-fwd_NotI/DmxA_Rev_HindIII. The fragment was digested with HindIII and NotI, cloned into pMAL-c6t, and sequenced.

pTP139 (plasmid for expression of His$_6$-*dmxA*$^{223-722, E626A}$): The *dmxA*$^{223-722, E626A}$ insert was amplified using pTP137[39] as DNA template and the primer pairs 3705 GAF1 forw NdeI/3705 E626A (−) and 3705 E626A (+)/3705 rev BamHI. To generate the full-length insert, an overlapping PCR using the two fragments as DNA templates and the primer pair 3705 GAF1 forw NdeI/3705 rev BamHI was performed. The fragment was digested with NdeI and BamHI, cloned into pET28a(+), and sequenced.

pMH117 (plasmid for expression of *malE-dmxA*$^{E626A (223-722, E626A)}$): The *dmxA*$^{223-722, E626A}$ insert was amplified using pTP139 as DNA template and the primer pair MalE-fwd_NotI/DmxA_Rev_HindIII. The fragment was digested with HindIII and NotI, cloned into pMAL-c6t, and sequenced.

pMP082 (plasmid for expression of His$_6$-*dmxA*$^{223-722, R615A}$): The *dmxA*$^{223-722, R615A}$ insert was amplified using pTP137[39] as DNA template and the primer pairs 3705 GAF1 forw NdeI/3705 R615A (−) and 3705 R615A (+)/3705 rev BamHI. To generate the full-length insert, an overlapping PCR using the two fragments as DNA templates and the primer pair 3705 GAF1 forw NdeI/3705 rev BamHI was performed. The fragment was digested with NdeI and BamHI, cloned into pET28a(+), and sequenced.

pMH116 (plasmid for expression of *malE-dmxA*$^{R615A (223-722, R615A)}$): The *dmxA*$^{223-722, R615A}$ insert was amplified using pMP082 as DNA template and the primer pair MalE-fwd_NotI/DmxA_Rev_HindIII. The fragment was digested with HindIII and NotI, cloned into pMAL-c6t, and sequenced.

pMH114 (plasmid for expression of *malE-dmxA*$^{GAF×2 (223-547)}$): The *dmxA*$^{223-547}$ insert was amplified using genomic DNA from *M. xanthus* DK1622 as DNA template and the primer pair MalE-fwd_NotI/DmxA_GAF2_rev_HindIII. The fragment was digested with HindIII and NotI, cloned into pMAL-c6t and sequenced.

pMH115 (plasmid for expression of *malE-dmxA*$^{GGDEF (546-722)}$): The *dmxA*$^{546-722}$ insert was amplified using genomic DNA from *M. xanthus* DK1622 as DNA template and the primer pair MalE-fwd_NotI/DmxA_GGDEFonly_Fwd_NotI. The fragment was digested with HindIII and NotI, cloned into pMAL-c6t, and sequenced.

pMAT74 (for in-frame deletion of *ftsZ*): up- and downstream fragments were amplified using genomic DNA from *M. xanthus* DK1622 as a DNA template and the primer pairs ftsZ-up EcoRI/ftsZ-overlapping reverse and ftsZ-overlapping forward/ftsZ-down HindIII, respectively. Subsequently, the up- and downstream fragments were used as a template for an overlapping PCR with the primer pair ftsZ-up EcoRI/ftsZ-down HindIII to generate the AD fragment. The AD fragment was digested with EcoRI and HindIII, cloned in pBJ114, and sequenced.

pMAT86 (plasmid for expression of *ftsZ* from the *MXAN_18-19* site under the control of the vanillate promoter): the *ftsZ* fragment was amplified using genomic DNA from *M. xanthus* DK1622 as DNA template and the primer pair ftsZ-start NdeI/ftsZ-stop KpnI. The fragment was digested with NdeI and KpnI, cloned into pMR3691, and sequenced.

pMH52 (plasmid containing an *M. xanthus* codon-optimized *miniTurbo-FLAG*) The *miniTurbo* sequence[92] was codon optimized for *M. xanthus*, synthesized with a N-terminal GGGS-linker and an C-terminal FLAG-tag and cloned into the pEX-k168.

pMP172 (plasmid for expression of *dmxA-miniTurbo-FLAG* under control of the *pilA* promoter from the *attB* site): The *dmxA* fragment was amplified with the primer pair 3705_PpilA forw/3705_rev no stop 1 from pMP092, and the *miniTurbo-FLAG* fragment was amplified with

the primer pair 3705_miniTurboID fw/Flag_rev HindIII from pMH52. Next, an overlapping PCR was performed using the previous PCR products and the primer pair 3705_PpilA forw/Flag_rev HindIII. The product was digested with XbaI and HindIII, cloned into pSW105, and sequenced.

pMH98 (plasmid for replacement of *ftsK* with *ftsK-miniTurbo-FLAG* in the native site): up- and downstream fragments were amplified using genomic DNA from *M. xanthus* DK1622 as DNA template and the primers FtsK_A_BamHI/FtsK_B_TID_OV and FtsK_C_TID_OV/FtsK_D_HindIII, respectively. The *miniTurbo-FLAG* fragment was amplified using pMH52 and the primer pair TID_fwd_FtsK_OV/TID_rev_FtsK_OV. To generate the full-length insert, an overlapping PCR using the three fragments as DNA templates and the primer pair FtsK_A_BamHI/FtsK_D_HindIII was performed. The fragment was digested with BamHI and HindIII, cloned into pBJ114, and sequenced.

pMH110 (plasmid for expression of *ftsK-miniTurbo-FLAG* from the *attB* site under the control of the *pilA* promoter): *ftsK* was amplified using genomic DNA from *M. xanthus* DK1622 as DNA template and the primer pair FtsK_fwd_XbaI_2/FtsK_B_TID_OV 2. The *miniTurbo-FLAG* insert was amplified from pMH98 using the primer pair TID_fwd_FtsK_OV/Flag_rev HindIII. To generate the full-length insert, an overlapping PCR using the two fragments as DNA templates and the primer pair FtsK_fwd_XbaI_2/Flag_rev HindIII was performed. The fragment was digested with XbaI and HindIII, cloned into pSW105, and sequenced.

pMH123 (plasmid for expression of codon-optimized cdGreen2 biosensor and mScarlet-I from the *attB* site under the control of the *pilA* promoter): The codon-optimized cdGreen2 biosensor and mScarlet-I full-length insert was amplified using pBBR15.2-2H12.D11opt-scarRef[55] as DNA template and the primer pair cdG-Sensor_fwd_XbaI/cdG-Sensor_rev_HindIII. The fragment was digested with XbaI and HindIII, cloned into pSW105, and sequenced.

### Reporting summary
Further information on research design is available in the Nature Portfolio Reporting Summary linked to this article.

## Data availability
The mass spectrometry proteomics data of whole cell proteomics and proximity labeling experiments have been deposited to the ProteomeXchange Consortium via the PRIDE partner repository with the dataset identifier PXD049046. The authors declare that all data supporting this study are available within the article, its Supplementary Information file, or in the Source Data file. All materials used in the study are available from the corresponding author. Source data are provided with this paper.

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

## Acknowledgements

We thank Sean Murray for helpful discussions and Dorota Skotnicka and Anke Treuner-Lange for strains, and the Research Core Unit Metabolomics at the Hannover Medical School for the assistance with measuring c-di-GMP levels. This work was supported by Deutsche Forschungsgemeinschaft (DFG, German Research Council) within the framework of the SFB987 "Microbial Diversity in Environmental Signal Response" (L.S.-A.), by the Max Planck Society (L.S.-A.), and by the Swiss National Science Foundation grant 310030_208107 (U.J.).

## Author contributions

The order of the two shared first authors is alphabetical and has no further meaning. Conceptualization: M.P.-B., M.H., L.S.-A.; Experimental work: M.P.-B., M.H., A.H., K.H., and T.G.; Visualization: M.P.-B. and M.H.; Software: M.H.; Formal analysis: M.P.-B., M.H., A.H. and T.G.; Resources: A.K. and U.J.; Funding acquisition: U.J. and L.S.-A.; Writing—original draft: M.P.-B., M.H. and L.S.-A.; Writing—review and editing: M.P.-B., M.H., A.K., U.J., T.G. and L.S.-A.

## Funding

## Competing interests

The authors declare no competing interests.
