## [Peer Review File · Nature Communications]

A deterministic, c-di-GMP-dependent program ensures the generation of phenotypically similar, symmetric daughter cells during cytokinesisReviewer #1 (Remarks to the Author):

This manuscript is on investigating how DmxA, a diguanylate cyclase (DGC), may function to regulate *Myxococcus xanthus* Type IV Pilus (T4P)-mediated motility. Previously, the same group showed that a *dmxA* insertion led to reduced T4P motility, elevated c-di-GMP levels and increased exopolysaccharide (EPS) production and no change or perhaps a slight increase in piliation. The *dmxA* insertion mutation was successfully complemented in the earlier study. Here a clean deletion was constructed and successfully complemented. The *dmxA* mutant showed similar motility phenotype on plate (macroscopic assay) as previously reported, except that a quantitative gliding defect was also noted in this study. The authors took tremendous efforts to try to understand the underlying mechanisms for the regulation of T4P-dependent motility by DmxA. They conclude from their observations that the DGC activity of DmxA is cell cycle-dependent. It regulates or promotes assembly of motility machinery at the nascent cell poles, leading to a more symmetric presence of the T4P machinery at the two new poles of the two newborn cells. More specifically, they propose that DmxA is recruited to the divisome prior to cytokinesis. This recruitment or localization leads to a transient activation of its DGC activity to result in a significant burst of c-di-GMP production. This burst, which lasts about 20 minutes before and after cytokinesis, is proposed to affect the correct localization of motility proteins through unknown mechanisms.

The authors are to be commended for their Herculean effort through extensive experimentation. Important and critical observations are presented in this manuscript. Their model is novel and intriguing. I believe that there is certainly some validity to the model they proposed.

One key problem lies in the inability to synchronize *M. xanthus* populations. As a result, the authors have to be highly selective in their data acquisition and analysis. In other words, the data sets that are relevant to cell cycle are quite small and very selective. It presents challenges for the argument that the data are unbiased and statistically significant. The results from static/snapshot images of cell populations, while quite large, are of questionable relevance to the key element of the model, which is the cell cycle-dependent regulation.

Specific comments:

1. L48, there is no specific references supporting the statement that c-di-GMP stimulates T4P-dependent motility.
2. L97, it is difficult to argue that the burst of c-di-GMP of such high magnitude that last 20 minutes in both the mother and daughter cells with uniformity only promotes protein localization or complex assembly at the new poles before or after cytokinesis. 20 minutes each way is a long time, especially considering that the resolution of time-lapse video is 10 minutes long in this case.
3. L116, preparative SEC columns shouldn't be used to estimate protein oligomeric state.
4. L134, how this intracellular concentration of c-di-GMP comes about is unclear.
5. L174, the higher bipolar piliation levels of *dmxA* mutants seem to argue against the lack or delay of T4PM at the new pole of a daughter cell. From the images that are shown, it almost seems the *dmxA* mutant are hyperpiliated. The inconsistency with higher c-di-GMP levels of the mutant in their previous publication should be addressed. It is curious that there is no result on EPS production in this manuscript (elevated in previous publication). While considering a model, perhaps keep in mind that there has been reports in the literature that the absence of T4P or gliding motility influence cell reversal by the other system.
6. L184-L196, what were the criteria for the selection of cells for Fig. 3b? Are they selected based on the presence of fluorescent clusters or on the observation of cell division events by phase contrast? The former would be highly biased. If it is the latter, indicate percentages of division events that didn't coincide with fluorescent clusters. I appreciate the data on the percentage of cells with constriction in Fig. 3A on the analysis of static images. This comment also applies to Fig. 4b.
7. L291, the statement on PiIT is contradictory to this group's own published results.
8. L296, how many cells were observed for Fig. 5a (Fig. S7 is missing)? The same question applies to Fig. 6a. This is important because the data in Fig. 5b (and Fig. 6b) cannot be interpreted easily in the context of cell cycle because they are static images of cells with exceedingly few in the process of cytokinesis.
9. L306, please explain the method for determining the statistical significance indicated in Fig. 5b, right panel. The data at the bottom right on PiIB were considered significantly different between WT and *dmxA* mutant. How? The same for Fig. 6b.

10. L380, the evidence for interactions between DmxA and FstK is not strong.
11. L423, the suggestion of the involvement of RomR as a downstream component of DmxA signaling is puzzling. There are proteins with c-di-GMP effector domains (PilZ, MshE, degenerate DGCs, etc) in *M. xanthus*.
12. Fig. 7, should there be a c-di-GMP concentration gradient in cells?

Reviewer #2 (Remarks to the Author):

The manuscript by Pérez-Burgos et al. revealed that DmxA, a diguanylate cyclase that was previously described as a regulator for type IV pilus-dependent motility, actually ensures two daughter cells to have mirror-symmetric localization of polarity regulators after division. Using a series of experiments, the authors proposed a model in which DmxA localizes to the division site and causes a burst of c-di-GMP, which "brings about" certain putative "landmark protein" that position polarity regulators to correct cell poles.

Although the experiments were well-executed and carefully analyzed, this manuscript has brought more questions than it answered. The reviewer is not convinced by the authors model and data interpretation.

My major questions:

1. The burst of c-di-GMP caused by DmxA is diffusive along the whole cell, which is easy to understand regarding the free diffusion of c-di-GMP. Then How can diffusive c-di-GMP cause proteins, such as PilQ, MglA, and MglB, or the putative "landmark proteins" to localize to asymmetric patterns? Simply saying that diffusive c-di-GMP could "bring about" landmark proteins to certain foci is far from enough to explain the reported results.
2. The manuscript proposed that in opposite to *C. crescentus* and *P. aeruginosa* that generate uneven c-di-GMP distribution between daughter cells, *M. xanthus* employs a mechanism to distribute c-di-GMP evenly. If DmxA does so by producing mirror-like c-di-GMP bursts in daughter cells, do the authors have any evidence that c-di-GMP concentrations are uneven in daughter cells in the absence of DmxA?
3. According to the manuscript, the authors didn't suspect that DmxA recruits polar regulators directly. If this is the case, there must be some proteins that fulfill such functions by responding to c-di-GMP. Do any such proteins seem work in the same pathway with DmxA?
4. The c-di-GMP bursts disappear rapidly after division, implying that a PDE that clears up c-di-GMP could be equally important for polarity setup. The authors also realized this possibility in L397-400. However, as the authors pointed out, none of the putative PDEs have been implicated in motility. How do the authors explain this?

Some minor questions:

1. L306-308. I had a hard time to understand this sentence. Doesn't "asymmetry" already mean lower fluorescence signal at one pole?
2. Fig. 1. It will be nice if panels a and c have matching colors. It's recommended to show the side chains of A and I sites.
3. Fig. 4d. Why not bleach cdGreen2? doing so will provide more information.

Reviewer #3 (Remarks to the Author):

The work by Burgos et al. shows that 1) of the 11 DGCs in *M. xanthus*, only DmxA localizes to the division site just prior to division, 2) c-di-GMP synthesis by this enzyme is detected around the time of localization, 3) in the absence of DmxA, T4 pilus components do not localize to the division pole. These conclusions are convincing, although the findings are not novel as far as developmental localization of c-di-GMP enzymes to cellular sites. The problematic part was the

writing, which was hard to follow in many places, and which over-sold and over-interpreted the findings. For example, in absence of data regarding the heterogeneity and noise in DmxA expression, the claim that the localization mechanism is designed to 'minimize phenotypic heterogeneity' is not grounded in reality. So also the claim that 'bursts of c-di-GMP' are designed to ensure 'equal and symmetric allocation of core T4PM proteins'. Also troubling were oxymoronic statements such as 'deterministic genetic programs' that are 'hardwired'. Overall, the multiple moving apparatuses and their parts made this manuscript a difficult read.

Major Comments

1. The Results section should have started with the main thrust of this paper - developmental regulation of DmxA localization. Instead, the reader has to wade through a deluge of purification and mutant activity data in Figs. 1 & S1 that distract from the main point. These data should perhaps be published separately.
2. The word 'deterministic' is used generously without defining what is meant. In biology, one understands determinism to mean a rigid process largely unaffected by environmental factors i.e. determined by heredity. In which case, isn't 'deterministic genetic program' an oxymoron? Doesn't 'hardwired' mean the exact same thing?
3. A straightforward interpretation of the DmxA localization is that local c-di-GMP production at the pole promotes assembly of the T4P/Gliding machinery. That this phenomenon is meant to 'minimize heterogeneity' is an overinterpretation.
4. Lines 71- 87 describe too many components for a non-Myxo reader to easily digest.
5. Fig. 2a, Lines 148-151. T4-P and Alg/Glt are two different modules of moving. So why is the phenotype of the *dmxA/gltB* double interpreted as a T4P defect?
6. Fig. 2a. The morphologies on 1.5 % agar in the first 3 panels don't look different to me. I also don't see a difference between *pilA* and *pilAdmxA* double that would allow the authors to conclude that *dmxA* affects gliding.
7. Lines 160-183 are not clearly written and the associated Fig. 2bc is incomprehensible. What I got from this is that in the absence of the reversal machinery Frz, cells can still reverse, which should have been stated right at the start. This should have been followed by FrzE-independent reversals seen in GAP mutants.
What are the steady state levels of DmxA across cells and what is the measurement of gene expression noise?
What happens if you overexpress DmxA in Fig. 2b?
In 2d, if 79% of *dmxA* mutants have T4P at one end, isn't that similar to the 82% at one end in WT?
8. Fig. 3b. The frequency distribution of cluster life-time and associated noise calculations should be reported.
Claims of a 'deterministic' event should be supported by measuring DmxA levels across multiple generations.
- Fig. 3c. The expt done with FtsZ should be repeated with FtsK for validating results of proximity labeling.
- Fig. 3d-e. In *E. coli*, the divisome is made of over 3 dozen proteins of which a dozen are essential. Why did the proximity labeling pick up only FtsK, whose enzymatic activity is needed largely to resolve chromosome dimers? There was no discussion about this. Why was FtsZ, which forms a dominant Z ring, not identified? Why was FtsB, which is co-transcribed with *dmxA* not identified? Why were simple pull-down experiments not done? Overall, the paper showcases all the latest technologies, ignoring straightforward tried-and-tested assays.
9. Line 220. This title doesn't make sense if the catalytically inactive E626A mutant is located at the constriction to the same extent as WT as shown in Fig. 3g. Also, why is WT DmxA localization not shown for comparison in this figure?

10. Fig. 4d and lines 271-276. The lines are EXTREMELY poorly written. I had to wrestle with the image for a long time before understanding the point of the experiment.

11. Lines 279-280. Can this not be said about any generic DGC as well?

12. Line 372. The authors invoke a 'burst' of c-di-GMP as ensuring 'the equal and symmetric allocation of core T4PM proteins and polarity proteins to the two daughters' here and elsewhere. However, a test of this argument would be that constitutive expression of DmxA would not ensure symmetric allocation. This experiment needs to be performed before leaping to the conclusion they do.

13. Line 378. If DmxA recruitment to the division site late during cytokinesis depends on the TMD of DmxA and divisome assembly, it does not necessarily suggest that the TMD interacts with the divisome. It could just as well be due to optimal activity of the full-length protein.

Minor comments

Line 106. How was the sequence analysis done? Any references?

Line 107. What is a GAF domain?

Reviewer #4 (Remarks to the Author):

REVIEWER COMMENTS

Reviewer #1 (Remarks to the Author):

This manuscript is on investigating how DmxA, a diguanylate cyclase (DGC), may function to regulate *Myxococcus xanthus* Type IV Pilus (T4P)-mediated motility. Previously, the same group showed that a *dmxA* insertion led to reduced T4P motility, elevated c-di-GMP levels and increased exopolysaccharide (EPS) production and no change or perhaps a slight increase in piliation. The *dmxA* insertion mutation was successfully complemented in the earlier study. Here a clean deletion was constructed and successfully complemented. The *dmxA* mutant showed similar motility phenotype on plate (macroscopic assay) as previously reported, except that a quantitative gliding defect was also noted in this study. The authors took tremendous efforts to try to understand the underlying mechanisms for the regulation of T4P-dependent motility by DmxA. They conclude from their observations that the DGC activity of DmxA is cell cycle-dependent. It regulates or promotes assembly of motility machinery at the nascent cell poles, leading to a more symmetric presence of the T4P machinery at the two new poles of the two newborn cells. More specifically, they propose that DmxA is recruited to the divisome prior to cytokinesis. This recruitment or localization leads to a transient activation of its DGC activity to result in a significant burst of c-di-GMP production. This burst, which lasts about 20 minutes before and after cytokinesis, is proposed to affect the correct localization of motility proteins through unknown mechanisms.

The authors are to be commended for their Herculean effort through extensive experimentation. Important and critical observations are presented in this manuscript. Their model is novel and intriguing. I believe that there is certainly some validity to the model they proposed. One key problem lies in the inability to synchronize *M. xanthus* populations. As a result, the authors have to be highly selective in their data acquisition and analysis. In other words, the data sets that are relevant to cell cycle are quite small and very selective. It presents challenges for the argument that the data are unbiased and statistically significant. The results from static/snapshot images of cell populations, while quite large, are of questionable relevance to the key element of the model, which is the cell cycle-dependent regulation.

Response: Thank you very much for your very constructive feedback and thoughtful comments. Regarding the “*data sets that are relevant to cell cycle are quite small and very selective*” we want to emphasize that in all time-lapse recordings, we initially watch cells by phase contrast microscopy. Subsequently, we randomly pick cells and check for localization of the fluorescently-tagged protein of interest. In doing so, we follow many cells starting at different stages of the cell cycle over the cell cycle (5-6hrs) and, therefore, we do not have to synchronize cells. So, we have a very comprehensive set of unbiased data describing protein localization over the cell cycle. These data clearly demonstrate that (1) DmxA-mVenus only localizes to the cell division site late during cytokinesis; (2) the c-di-GMP level increases late during cytokinesis; and, (3) PilQ and RomR incorporation and allocation at the nascent cell poles and new cell poles of the two daughters are symmetric in WT cells and this symmetry depends on DmxA. In other words, our data from time-lapse recordings are not selective and not biased and lead to the three mentioned conclusions. To bolster these conclusions, we also include snapshot analyses of thousands of cells. These analyses are also not selective or biased. Importantly, they support the conclusions based on the time-lapse recordings. Because the localization of DmxA and the c-di-GMP burst late during cytokinesis are entirely novel, we consider it very important to have two different experimental approaches to document these findings. So, we disagree with the statement that the snapshot data are of “*questionable relevance*”. Finally, we would like to emphasize that we have taken great care that our sample

sets are sufficiently large to make statements about statistically significant differences. To further clarify that our data are not “*small and very selective*” we have taken the following actions. (i) amended all figures and/or figure legends involving time-lapse recordings and detailed how many cells were analysed and that the images represent representative cells; (ii) changed the presentation of the quantitative analyses of DmxA localization and the c-di-GMP level over the cell cycle in Fig. 3b and Fig. 4b, respectively; (iii) included new analyses in Fig. 5a and Fig. 6a with detailed quantitative analyses of the localization of PilQ and RomR during and after division, (iv) amended the main text (line 202, 275, 278, 301, 323, 326, 354, 358) to also here clearly state how many cells were analysed in time-lapse recordings. Importantly, the new quantitative analyses in Fig. 5a and Fig. 6a fully support our previous conclusions. In total, the new quantifications of signals during cell division combined with the quantification of populations in snapshots reflect the effect that DmxA has on the motility and polarity proteins during the complete cell cycle.

Specific comments:

1. L48, there is no specific references supporting the statement that c-di-GMP stimulates T4P-dependent motility.

Response: Thanks for pointing this out. We have now added the original papers in which c-di-GMP binding to effectors was shown to stimulate surface adhesion and T4P-dependent surface motility (line 50).

2. L97, it is difficult to argue that the burst of c-di-GMP of such high magnitude that last 20 minutes in both the mother and daughter cells with uniformity only promotes protein localization or complex assembly at the new poles before or after cytokinesis. 20 minutes each way is a long time, especially considering that the resolution of time-lapse video is 10 minutes long in this case.

Response: We apologize for not being sufficiently clear. To increase clarity and describe that (1) DmxA begins to localize at the division site ~20 min prior to completion of cytokinesis, followed by disintegration of the DmxA-mVenus cluster upon completion of cytokinesis, and (2) the DmxA-dependent increase in c-di-GMP starts ~20 min prior to completion of cytokinesis, followed by its degradation upon completion of cytokinesis, we amended the main text in line 199-203 to “*Indeed, we observed by time-lapse microscopy in which cells were followed over the cell cycle (Fig. 3b) that (i) DmxA-mVenus formed a cluster at the division site in all division events but only after initiation of cytokinesis (n=100)*” and line 274-278 to “*Next, we used time-lapse microscopy to follow the cdGreen2 signal dynamics over time. The analysis of $\Delta 10$ (n=130) and WT (n=73) cells revealed that the cdGreen2 signal specifically increased dramatically in all cells shortly before completion of cytokinesis and then decreased rapidly in the two daughters, which had equal levels of cdGreen2 fluorescence (Fig. 4b; S6d). By contrast, Δ dmxA cells (n=67) completely lacked the transient increase in c-di-GMP (Fig. 4c)*” and also modified Fig. 3b and Fig. 4b. Specifically, in these two figures, we changed the visualization of the “time of arrival” of DmxA at the division site relative to completion of cytokinesis and the DmxA-dependent increase in c-di-GMP relative to completion of cytokinesis.

Moreover, we amended the Discussion in line 470-474 to clarify that our data support that DmxA ensures the incorporation of a landmark(s) at the nascent and new poles that may only be transiently active. We also included two examples where c-di-GMP promotes the transient localization of polar landmark proteins that are also c-di-GMP effectors.

3. L116, preparative SEC columns shouldn't be used to guesstimate protein oligomeric state.

Response: Point well taken! We toned down our conclusion in line 112-114 and emphasized that these conclusions are based on estimations. Nevertheless, because we are comparing large mass shifts in variants of the same protein, we believe that this comparison allows us to make inferences on oligomeric states. Furthermore, our SEC results are supported by the AlphaFold model, which supports that DmxA dimerizes and that the main interaction interface for the dimer is provided by the GAF domain-containing region.

4. L134, how this intracellular concentration of c-di-GMP comes about is unclear.

Response: The c-di-GMP levels was measured in a previous study and was published in Skotnicka et al. 2020. We have rephrased the sentence (line 131) to make clear that it is based on previous measurements.

5. L174, the higher bipolar piliation levels of dmxA mutants seem to argue against the lack or delay of T4PM at the new pole of a daughter cell. From the images that are shown, it almost seems the dmxA mutant are hyperpiliated. The inconsistency with higher c-di-GMP levels of the mutant in their previous publication should be addressed. It is curious that there is no result on EPS production in this manuscript (elevated in previous publication). While considering a model, perhaps keep in mind that there has been reports in the literature that the absence of T4P or gliding motility influence cell reversal by the other system.

Response: We apologize for not being sufficiently clear.

In the electron microscopy experiments, we assay the piliation patterns. To address the reviewer's comment about the level of piliation, we have included a new experiment in which we address the level of piliation in WT and $\Delta dmxA$ cells using a shear-off assay. This experiment shows that the $\Delta dmxA$ mutant is indeed slightly hyper-piliated (Fig. S2b and described in line 178-181). For completeness, we additionally added a representative image of a WT and a $\Delta dmxA$ cell with unipolar T4P in Fig. S2a.

We also added a new experiment in which we assess the level of EPS production in WT and $\Delta dmxA$ cells in Fig. S2c and line 181-184. This experiment shows, in agreement with previously published results, that the $\Delta dmxA$ mutant has a slightly increased EPS production compared to WT.

Concerning the previously measured c-di-GMP level, we have included a brief explanation in line 295-297 that we believe this difference is caused by using two different mutants.

We agree with the reviewer that intuitively the faulty assembly of the T4PM would result in more unipolarly piliated cells. Because we observed the bipolarly piliated cells, we went ahead and analysed the localization of proteins of the polarity module. Based on the observations that lack of DmxA also affects their localization, we suggest in line 448-450 "*that the aberrant T4P-dependent motility behaviour in the absence of DmxA is the result of dual defects, i.e. the defects in the polar incorporation of the core T4PM and in the localization of polarity proteins*".

We very much appreciate the comment that lack of one motility system could possibly affect reversals in the remaining system. To streamline and clarify the description of the motility defects caused by lack of DmxA, we edited the main text in line 144-153 and line 158-173.

Specifically, the double mutants $\Delta dmxA\Delta gltB$ and $\Delta dmxA\Delta pilA$ both have defects in the remaining motility system. Therefore, we can solidly conclude that DmxA impacts both motility systems. Similarly, the finding that proteins of the polarity module are mislocalized in the absence of DmxA supports that both motility systems are affected by lack of DmxA.

6. L184-L196, what were the criteria for the selection of cells for Fig. 3b? Are they selected based on the presence of fluorescent clusters or on the observation of cell division events by phase contrast? The former would be highly biased. If it is the latter, indicate percentages of division events that didn't coincide with fluorescent clusters. I appreciate the data on the percentage of cells with constriction in Fig. 3A on the analysis of static images. This comment also applies to Fig. 4b.

Response: As outlined above in our response to the reviewer's general comments, in all time-lapse recordings, we initially watch cells by phase contrast microscopy. Subsequently, we check for localization of the fluorescently-tagged protein of interest. In doing so, we follow many cells starting at different stages of the cell cycle over the cell cycle. Therefore, we have a very comprehensive set of unbiased data describing protein localization over the cell cycle. To help clarify this point, we have modified the text in line 199-201 to make clear that all cells had a DmxA-mVenus cluster at mid-cell during division (Fig. 2b). Similarly, we have modified the text in line 273-278 to make clear all WT cells had an increased cdGreen2 signal during division (Fig. 4b) while $\Delta dmxA$ cells completely lacked this transient increase (Fig. 4c).

7. L291, the statement on PiIT is contradictory to this group's own published results.

Response: We apologize for not being sufficiently clear. In our initial analyses of PiIT localization (Bulyha et al., 2009), we used a strain overexpressing a PiIT-YFP fusion and found that the fusion localized in a bipolar asymmetric pattern. In the meantime, we use an mCherry-PiIT fusion that is expressed at native levels either ectopically (Potapova et al. 2020) or from the native locus (Oklictschek et al. 2024). This fusion localizes bipolarly. We have now included these two references for the description of the mCherry-PiIT localization pattern (line 318).

8. L296, how many cells were observed for Fig. 5a (Fig. S7 is missing)? The same question applies to Fig. 6a. This is important because the data in Fig. 5b (and Fig. 6b) cannot be interpreted easily in the context of cell cycle because they are static images of cells with exceedingly few in the process of cytokinesis.

Response: We apologize for not being sufficiently clear. We have now included in the main text (line 322, 325, 353, 357) and in Fig. 5a and Fig. 6a how many cells were analysed during cytokinesis. Moreover, we included new analyses in Fig. 5a and Fig. 6a in which we quantify the asymmetry index of the predivisional cell before and the two daughters after division in the WT and in the $\Delta dmxA$ mutant. These analyses clearly show that there is defect in the incorporation and allocation of PilQ-sfGFP and RomR-mCherry at the nascent poles and new poles.

Fig. S7 was a mistake. We apologize.

9. L306, please explain the method for determining the statistical significance indicated in Fig. 5b, right panel. The data at the bottom right on PilB were considered significantly different between WT and $dmxA$ mutant. How? The same for Fig. 6b.

Response: To compare the distribution of the asymmetry index in the different strains in Fig. 5b and Fig. 6b, we used the Mann-Whitney test. In the plot in Fig. 5b for PilB, we find a P -value =

0.0007. Specifically, the distribution of the asymmetry index of PilB in the $\Delta dmxA$ mutant extends more into the unipolar direction than in WT. Statistical methods used are included in the figure legends as well as in the Method section line 755-761.

10. L380, the evidence for interactions between DmxA and FtsK is not strong.

Response: Proximity labelling is generally taken as evidence that two proteins are within a distance of 10-20nm. But proximity labelling does not provide direct evidence that two proteins interact directly. Therefore, throughout the text, we take great care to state that DmxA interacts with protein(s) of the divisome, and is thereby recruited to the division site late during cytokinesis, and that FtsK is a potential interaction partner of DmxA. It is clear that we have not shown that DmxA and FtsK directly interact, only that the divisome is important for DmxA recruitment. To extend on this point, we have expanded the discussion of the proximity labelling experiments in line 409-415.

11. L423, the suggestion of the involvement of RomR as a downstream component of DmxA signaling is puzzling. There are proteins with c-di-GMP effector domains (PilZ, MshE, degenerate DGCs, etc) in *M. xanthus*.

Response: We apologize for not being sufficiently clear. To clarify, we state neither that RomR is a c-di-GMP receptor nor that RomR directly interacts with DmxA. We speculate in the Discussion (line 453-473) that the c-di-GMP burst generated by DmxA is important for the recruitment of a landmark(s) to the nascent and new poles and that this landmark(s) helps to recruit PilQ and RomR. As stated in line 473-475, this landmark(s) remains to be identified. We expanded this part of the Discussion (line 469-473) to include two examples from other bacteria in which a c-di-GMP binding protein functions as a landmark protein. We have previously identified c-di-GMP binding proteins in *M. xanthus* that affect motility. These proteins include the PilZ-domain proteins PixA and PixB and the histidine protein kinase SgmT. However, for none of these proteins, was c-di-GMP binding important for their function in motility. Therefore, we did not include a discussion of these proteins in the Discussion.

12. Fig. 7, should there be a c-di-GMP concentration gradient in cells?

Response: Good point! We modified Figure 7 and removed the c-di-GMP gradient because it is unclear how a small diffusible molecule like c-di-GMP would be able to form a gradient in the cytoplasm.

Reviewer #2 (Remarks to the Author):

The manuscript by Pérez-Burgos et al. revealed that DmxA, a diguanylate cyclase that was previously described as a regulator for type IV pilus-dependent motility, actually ensures two daughter cells to have mirror-symmetric localization of polarity regulators after division. Using a series of experiments, the authors proposed a model in which DmxA localizes to the division site and causes a burst of c-di-GMP, which "brings about" certain putative "landmark protein" that position polarity regulators to correct cell poles.

Although the experiments were well-executed and carefully analyzed, this manuscript has brought more questions than it answered. The reviewer is not convinced by the authors model and data interpretation.

My major questions:

1. The burst of c-di-GMP caused by DmxA is diffusive along the whole cell, which is easy to understand regarding the free diffusion of c-di-GMP. Then How can diffusive c-di-GMP cause proteins, such as PilQ, MglA, and MglB, or the putative "landmark proteins" to localize to asymmetric patterns? Simply saying that diffusive c-di-GMP could "bring about" landmark proteins to certain foci is far from enough to explain the reported results.

Response: We speculate in the Discussion (line 453-473) that DmxA, when localized to mid-cell during cytokinesis, generates a burst of c-di-GMP and that this burst is important for the recruitment of a landmark(s) to the nascent and new poles and that this landmark(s) helps to recruit PilQ and RomR. As stated in line 473-475, this landmark(s) remains to be identified. We expanded this part of the Discussion (line 469-473) to include two examples from other bacteria in which a c-di-GMP binding protein functions as a landmark protein.

2. The manuscript proposed that in opposite to *C. crescentus* and *P. aeruginosa* that generate uneven c-di-GMP distribution between daughter cells, *M. xanthus* employs a mechanism to distribute c-di-GMP evenly. If DmxA does so by producing mirror-like c-di-GMP bursts in daughter cells, do the authors have any evidence that c-di-GMP concentrations are uneven in daughter cells in the absence of DmxA?

Response: As described in line 199-203 and shown in Fig. 3a-b, DmxA is explicitly recruited to the division site late during cytokinesis, and released upon completion of cytokinesis. As described in line 273-270 and shown in Fig. 4a-c, during this brief period of the cell cycle, its DGC activity is switched on in the predivisional cell resulting in a dramatic but transient increase in the c-di-GMP concentration. Upon completion of cytokinesis, the c-di-GMP level rapidly decreases. Thus, DmxA produces a c-di-GMP burst in the predivisional cell that is rapidly removed upon completion of cytokinesis. In the absence of DmxA, this brief burst in c-di-GMP is absent. According to the measurements using the cdGreen2 biosensor in a mutant lacking DmxA (Fig. 4a, c), the two daughters appear equal in terms of c-di-GMP level. Importantly, these daughters derive from a predivisional cell that was not exposed to the c-di-GMP burst and, therefore, they have defects in the polar incorporation and allocation to the daughters of proteins of the T4PM and the polarity module.

3. According to the manuscript, the authors didn't suspect that DmxA recruits polar regulators directly. If this is the case, there must be some proteins that fulfill such functions by responding to c-di-GMP. Do any such proteins seem work in the same pathway with DmxA?

Response: We apologize for not being sufficiently clear. We speculate in the Discussion (line 453-473) that the c-di-GMP burst generated by DmxA is important for the recruitment of a landmark(s) to the nascent and new poles and that this landmark(s) helps to recruit PilQ and RomR. As stated in line 473-475, this landmark(s) remains to be identified. We expanded this part of the Discussion (line 469-473) to include two examples from other bacteria in which a c-di-GMP binding protein functions as a landmark protein. We have previously identified c-di-GMP effectors in *M. xanthus* that affect motility. These effectors include the PilZ-domain proteins PixA and PixB and the histidine protein kinase SgmT. However, for none of these proteins, was c-di-GMP binding important for their function in motility. Therefore, we did not include a discussion of these proteins in the Discussion. Clearly, an important goal for the future will be to identify the c-di-GMP binding effector(s) involved in the response to DmxA-generated c-di-GMP.

4. The c-di-GMP bursts disappear rapidly after division, implying that a PDE that clears up c-di-GMP could be equally important for polarity setup. The authors also realized this possibility in

L397-400. However, as the authors pointed out, none of the putative PDEs have been implicated in motility. How do the authors explain this?

Response: According to our RNAseq data (Kuzmich et al. 2022), the six predicted PDEs of *M. xanthus* are expressed during growth. However, based on single gene deletions, none of them are implicated in motility, suggesting that they may function redundantly to cause the decrease in c-di-GMP upon completion of cytokinesis. We would like to add that, generally, it can be challenging to identify the components of c-di-GMP signalling networks. For instance, in *E. coli* four DGCs together with one master PDE fine-tune and modulate the c-di-GMP level to regulate motility. Single deletions of the genes for the four DGCs only lead to weak phenotypes. Along the same lines, in *P. aeruginosa*, the DGC that is responsible for the burst in c-di-GMP upon surface contact has not been identified. Altogether, at this point in time, we would like to keep the discussion about which PDE(s) are involved in the rapid degradation of DmxA-generated c-di-GMP short and with the comment that in the future, it will be interesting to identify the involved PDE(s) (line 434-436).

Some minor questions:

1. L306-308. I had a hard time to understand this sentence. Doesn't "asymmetry" already mean lower fluorescence signal at one pole?

Response: An increased asymmetry can be caused by one pole accumulating more protein than in WT or by one of the poles accumulating less protein than in WT. Therefore, we believe that it is important to include the statement (line 356-358) that "The shifts toward asymmetry were primarily caused by many $\Delta dmxA$ cells having no or a strongly decreased fluorescence signal at the pole with the lowest fluorescence (Fig. 5b).".

2. Fig. 1. It will be nice if panels a and c have matching colors. It's recommended to show the side chains of A and I sites.

Response: Thanks & done!

3. Fig. 4d. Why not bleach cdGreen2? doing so will provide more information.

Response: We used cdGreen2 to follow whether DmxA is active during the experiment. Therefore, we decided to bleach mScarlet-I to test for protein diffusion. Of note, mScarlet-I (26.6 kDa) is smaller in size than cdGreen2 (48.3 kDa). We rewrote lines 299-304 to clarify the general experimental procedure and why we bleached the mScarlet-I signal and not cdGreen2.

Reviewer #3 (Remarks to the Author):

The work by Burgos et al. shows that 1) of the 11 DGCs in *M. xanthus*, only DmxA localizes to the division site just prior to division, 2) c-di-GMP synthesis by this enzyme is detected around the time of localization, 3) in the absence of DmxA, T4 pilus components do not localize to the division pole. These conclusions are convincing, although the findings are not novel as far as developmental localization of c-di-GMP enzymes to cellular sites. The problematic part was the writing, which was hard to follow in many places, and which over-sold and over-interpreted the findings. For example, in absence of data regarding the heterogeneity and noise in DmxA expression, the claim that the localization mechanism is designed to 'minimize phenotypic heterogeneity' is not grounded in reality. So also the claim that 'bursts of c-di-GMP' are designed to ensure 'equal and symmetric allocation of core T4PM proteins'. Also troubling were oxymoronic statements such as 'deterministic genetic programs' that are 'hardwired'. Overall, the multiple moving apparatuses and their parts made this manuscript a difficult read.

Response: We thank the reviewer for the very constructive feedback. To make the text more accessible, we have modified it throughout.

We strongly disagree with the reviewer regarding the comment that our results are not novel. As described in line 491-494, in *C. crescentus*, *P. aeruginosa* and *S. putrefaciens*, c-di-GMP mediates the generation of phenotypically distinct daughter cells during cell division. This depends on the asymmetric deployment of c-di-GMP metabolizing enzyme(s) to the daughters during cell division, i.e. either the relevant DGC and PDE localize to opposite cell poles or a PDE localizes unipolarly. To the best of our knowledge, our results provide the first example where c-di-GMP mediates the generation of equal daughter cells and this involves placing a DGC at mid-cell of the pre-divisional cell.

Major Comments

1. The Results section should have started with the main thrust of this paper - developmental regulation of DmxA localization. Instead, the reader has to wade through a deluge of purification and mutant activity data in Figs. 1 & S1 that distract from the main point. These data should perhaps be published separately.

Response: We considered this suggestion but we have decided to keep the original structure because our *in vivo* analyses build logically on the *in vitro* analysis of the biochemical properties of DmxA.

2. The word 'deterministic' is used generously without defining what is meant. In biology, one understands determinism to mean a rigid process largely unaffected by environmental factors i.e. determined by heredity. In which case, isn't 'deterministic genetic program' an oxymoron? Doesn't 'hardwired' mean the exact same thing?

Response: Thanks for pointing this out to us. We have modified the text throughout to replace "deterministic, genetic program" with a "deterministic program" that results in the formation of symmetric daughters during cytokinesis. The determinism comes from this program being hardwired into the cell cycle. Therefore, we have kept "hardwired" in the text.

3. A straightforward interpretation of the DmxA localization is that local c-di-GMP production at the pole promotes assembly of the T4P/Gliding machinery. That this phenomenon is meant to 'minimize heterogeneity' is an overinterpretation.

Response: Thanks for making us think carefully about this. Because the c-di-GMP pool generated by DmxA is sensed by the cdGreen2 biosensor, we infer that DmxA contributes to a global pool of c-di-GMP. In line 478-480, we have included a brief discussion regarding local/global pools of c-di-GMP. We observe that lack of DmxA enzymatic activity and, therefore, the transient burst in c-di-GMP during cytokinesis result in the asymmetric allocation of T4PM and polarity proteins to the two daughters. In other words, the daughters are phenotypically different. Therefore, we conclude that DmxA enzyme activity and c-di-GMP ensure the formation of phenotypically similar cells, i.e. minimizes heterogeneity.

4. Lines 71-87 describe too many components for a non-Myxo reader to easily digest.

Response: We apologize and have rewritten the Introduction in line 67-82 to simplify and make it more accessible to non-Myxo readers.

5. Fig. 2a, Lines 148-151. T4-P and Alg/Glt are two different modules of moving. So why is the phenotype of the *dmxA/gltB* double interpreted as a T4P defect?

Response: We apologize for not being sufficiently clear. We have edited this section (line 144-157) to increase clarity. To address the specific question of the reviewer: The $\Delta gltB$ mutant can only move by means of T4P-dependent motility. Therefore, to clearly distinguish whether the $\Delta dmxA$ mutant has a defect in T4P-dependent motility, we compare the $\Delta dmxA\Delta gltB$ double mutant to the $\Delta gltB$ mutant.

6. Fig. 2a. The morphologies on 1.5 % agar in the first 3 panels don't look different to me. I also don't see a difference between pilA and pilAdmxA double that would allow the authors to conclude that dmxA affects gliding.

Response: Following the previous comment, we have edited the text to clarify the results of the motility assays (lines 144-157). We appreciate that for non-myxo readers it can be difficult to appreciate the differences between the different strains. For that reason, we measured the colony expansion and analysed for significant differences using the Student's t-test.

7. Lines 160-183 are not clearly written and the associated Fig. 2bc is incomprehensible. What I got from this is that in the absence of the reversal machinery Frz, cells can still reverse, which should have been stated right at the start. This should have been followed by FrzE-independent reversals seen in GAP mutants.

Response: We apologize for not being sufficiently clear. We rewrote this text to increase clarity (line 158-173).

What are the steady state levels of DmxA across cells and what is the measurement of gene expression noise?

Response: To address the reviewer's point, we included a new detailed analysis of DmxA-mVenus fluorescence. Specifically, we followed a previously established procedure (Ramm et al., 2023; Carreira et al., 2020) and determined the steady state levels of DmxA-mVenus in a mixed population of *M. xanthus* cells in which *dmxA-mVenus* is expressed from the native site. As described in line 206-213 and shown in Fig. S3a, we measured the total cellular DmxA-mVenus fluorescence in >1000 cells and then normalized the fluorescence intensities for cell area to obtain a metric for the cellular DmxA-mVenus concentration. These analyses show that all cells accumulate DmxA-mVenus and that the cellular DmxA-mVenus concentration follows a unimodal distribution. Consistently, and as described in line 213-215 & shown in Fig. S3b, based on time-lapse recordings, the normalized fluorescence intensities of DmxA-mVenus remained constant before, during and after cytokinesis. Based on these findings we infer (line 215-218) that "*the steady state level of DmxA-mVenus is constant over the cell cycle, strongly suggesting that localization of DmxA-mVenus to the division site, but not accumulation of DmxA, is cell cycle regulated*". In other words, the cellular DmxA-mVenus concentration does not follow a bimodal distribution. This constant cellular DmxA-mVenus concentration is determined by its synthesis and degradation. At this point, the important conclusion is that the cellular DmxA-mVenus concentration is constant and, therefore, we have determined neither the DmxA-mVenus rate of synthesis nor its rate of degradation and nor gene expression noise.

Supporting that DmxA accumulation is not cell cycle regulated, and as described in line 223-227, in cells treated with cephalixin for one generation, the frequency of cells with a constriction and a DmxA-mVenus cluster at mid-cell had significantly increased (Fig. 3a). However, DmxA-mVenus accumulated at the same level in cephalixin-treated and untreated cells (Fig. S4a). Similarly, and as described in line 229-233, cells depleted of FtsZ had neither constrictions nor

DmxA-mVenus clusters at mid-cell (Fig. 3c), despite the protein accumulating at a slightly higher level than in untreated cells (Fig. S4b).

Based on these three lines of experiments and analyses, we infer that the steady state level of DmxA is constant and not cell cycle-regulated (line 422-424).

What happens if you overexpress DmxA in Fig. 2b?

Response: We generated a strain overexpressing DmxA-mVenus. The protein levels of the overexpressing strain are significantly higher compared to a strain in which DmxA-mVenus is expressed from the native site. This strain also only forms clusters in ~5% of cells and preliminary data suggest that DmxA overexpression does not interfere with motility. We decided not to include the data for this strain because it is unclear to us what we learn from these experiments.

In 2d, if 79% of *dmxA* mutants have T4P at one end, isn't that similar to the 82% at one end in WT?

Response: We apologize for not being sufficiently clear. In the electron microscopy experiments, we assay the piliation patterns. To address the reviewer's comment about the level of piliation, we have included a new experiment in which we address the level of piliation in WT and $\Delta dmxA$ cells using a shear-off assay. This experiment shows that the $\Delta dmxA$ mutant is indeed slightly hyper-piliated (Fig. S2b & described in line 178-181). For completeness, we additionally added a representative image of a WT and a $\Delta dmxA$ cell with unipolar T4P in Fig. S2a.

8. Fig. 3b. The frequency distribution of cluster life-time and associated noise calculations should be reported.

Claims of a 'deterministic' event should be supported by measuring DmxA levels across multiple generations.

Response: We followed the reviewer's suggestion and plotted the frequency distribution of cluster appearance before completion of cytokinesis and cluster life-time in Fig. 3b. In addition, we modified the text in line 199-203 to emphasize that the DmxA-cluster appears in every division event in all cells observed. Because the DmxA cluster is visible in every division event, we did not analyse DmxA-mVenus signal over different generations. Please also refer to our answer to question 7.

Fig. 3c. The expt done with FtsZ should be repeated with FtsK for validating results of proximity labeling.

Response: Proximity labelling is generally taken as evidence that two proteins are within a distance of 10-20nm. But proximity labelling does not provide direct evidence that two proteins interact directly. Therefore, throughout the text, we take great care to state that DmxA interacts with protein(s) of the divisome, and is thereby recruited to the division site late during cytokinesis, and that FtsK is a potential interaction partner of DmxA. It is clear that we have not shown that DmxA and FtsK directly interact, only that the divisome is important for DmxA recruitment to the cell division site. To elaborate on this point, we have expanded the discussion of the proximity labelling experiments in line 409-415. Specifically, FtsZ is at mid-cell in ~50% of the cells (Treuner-Lange et al. 2012), FtsK in ~15% of the cells (Schumacher et al. 2017) and DmxA only in ~5% of cells, indicating that FtsK may not directly recruit DmxA. Interestingly, FtsK

is important for recruiting the late division proteins FtsQ, -L and -B, which are bitopic membrane proteins (Cameron & Margolin 2023), and DmxA is encoded in an operon with FtsB (Fig. S3d-e). Therefore, it remains a possibility that DmxA could be recruited to the division site by these protein(s). Of note, FtsQ, -L and -B only have short cytoplasmic tails, making them difficult targets for proximity labelling. Moreover, in proximity labelling experiments, proteins need to have Lys residue(s) appropriately localized in order to be labelled and for instance the 10 kDa FtsB protein is predicted to have 6aa residues in the cytoplasm, only one of which is a Lys residue.

Nevertheless, we have indeed tried to generate FtsK and FtsB depletion strains. However, we were unable to generate tightly regulated constructs in which we could completely shut-off their synthesis. Therefore, it was not possible for us to do the suggested experiment. Regardless, the important take home message of this experiment is that DmxA is recruited to the division site by the divisome.

Fig. 3d-e. In *E. coli*, the divisome is made of over 3 dozen proteins of which a dozen are essential. Why did the proximity labeling pick up only FtsK, whose enzymatic activity is needed largely to resolve chromosome dimers? There was no discussion about this. Why was FtsZ, which forms a dominant Z ring, not identified? Why was FtsB, which is co-transcribed with *dmxA* not identified? Why were simple pull-down experiments not done? Overall, the paper showcases all the latest technologies, ignoring straightforward tried-and-tested assays.

Response: Please see our answer to the previous question.

We would also like to add that FtsK is not only important for recruiting the FtsQLB complex to the divisome (Cameron & Margolin, 2023) but also for the XerCD recombinase (Castillo *et al.* 2017). Therefore, we were not surprised to find FtsK (which is also a large protein with a large cytoplasmic domain) in our proximity labelling experiments. We would also like to add that in our experience, pull-down experiments with membrane proteins such as DmxA and many of the divisome proteins are not straightforward. Moreover, if interactions only occur for a brief period of the cell cycle (e.g. 20 min out of 300 min), pull-down experiments are even more complicated. Regardless, the important take home message of this experiment is that DmxA is recruited to the division site by the divisome.

9. Line220. This title doesn't make sense if the catalytically inactive E626A mutant is located at the constriction to the same extent as WT as shown in Fig. 3g. Also, why is WT DmxA localization not shown for comparison in this figure?

Response: Thank you for pointing this out. We have modified the title to "DmxA function depends on DGC activity and DmxA is recruited to the division site by the TMD" (line 242-243). And, we included the image of the WT in panel 3g.

10. Fig. 4d and lines 271-276. The lines are EXTREMELY poorly written. I had to wrestle with the image for a long time before understanding the point of the experiment.

Response: We very much apologize. We rewrote line 297-304 to clarify the experimental procedure and its interpretation.

11. Lines 279-280. Can this not be said about any generic DGC as well?

Response: We deleted this comment (line 307).

12. Line 372. The authors invoke a 'burst' of c-di-GMP as ensuring 'the equal and symmetric allocation of core T4PM proteins and polarity proteins to the two daughters' here and elsewhere. However, a test of this argument would be that constitutive expression of DmxA would not ensure symmetric allocation. This experiment needs to be performed before leaping to the conclusion they do.

Response: As detailed in our response to the reviewer's question 7, we included new analyses to determine the cellular concentration of DmxA-mVenus. These analyses document that the cellular concentration of DmxA-mVenus is constant throughout the cell cycle. In other words, DmxA-mVenus accumulates constitutively resulting in a constant cellular concentration over the cell cycle. And, what is being regulated in a cell cycle-dependent manner is the localization and activation of DmxA-mVenus at the cell division site. Therefore, we disagree with the reviewer that constitutive expression of DmxA would not ensure symmetric allocation.

13. Line 378. If DmxA recruitment to the division site late during cytokinesis depends on the TMD of DmxA and divisome assembly, it does not necessarily suggest that the TMD interacts with the divisome. It could just as well be due to optimal activity of the full-length protein.

Response: Because the TMD is required and sufficient for DmxA-mVenus localization, we speculate in line 406-407 that the TMD interacts with the divisome. We agree that we cannot rule out that the TMD is also important for DmxA activity. Therefore we state in the text "Although we cannot rule out that DmxA^{ΔTMD} is less active than full-length DmxA" in line 420-422.

Minor comments

Line 106. How was the sequence analysis done? Any references?

Response: The description of the analysis is included in the Methods (line 736-737).

Line 107. What is a GAF domain?

Response: We added in line 102-103 that the GAF domain is named after the proteins in which it was initially identified: cGMP-specific phosphodiesterases, adenylyl cyclases and FhlA.

Reviewer #4 (Remarks to the Author):

Response: Thank you very much!

Reviewer #1 (Remarks to the Author):

I regret to reiterate that my primary concerns persist. Firstly, there is apprehension regarding the selective and limited nature of the datasets pertaining to the cell division cycle. The wording (L196-203) suggests that the cells analyzed for DmxA localization (Fig. 3a) were chosen based on fluorescent signals. Additionally, for the localization of PilQ and RomR, about 30 cells in total were analyzed for both the WT and the mutant (Figs. 5a & 6a). Of particular concern is the former (DmxA) as it is pivotal to the model.

Second, the model's validity remains in question as the authors have not addressed concerns raised by multiple reviewers. More specifically, the points #2, #5, #6, #8, #11 and #12 raised by this reviewer are all concerns related to one aspect or another of the proposed model. The summary by Reviewer #2 concluded that the "reviewer is not convinced by the authors model and data interpretation". Essentially all major points raised by this reviewer focused on why this conclusion was reached. The review by Reviewer #3 stated that the "problematic part was the writing..., which over-sold and overinterpreted the findings". It further states that "the claim that localization mechanisms... is not grounded in reality. So also the claim that.....". These concerns should not be overlooked.

The following are comments on their rebuttal to selected points this reviewer raised previously.

#1. None of the references indicated a stimulation of motility, only adhesion.

#3. There is no calibration with MW standards of a prep SEC column. The elution may suggest a change in oligomeric state, but not dimer vs monomer or pentamer vs hexamer.

#4. This goes back to a 2016 publication that measured c-di-GMP at pmol per mg protein. There is no explanation provided how this unit was converted to micromolar, which requires the conversion of protein to volume.

#7. The binary statement on PilB & PilT (L318) localization appears misleading. That is, it is inaccurate to state that "PilT localizes bipolarly, while PilB almost exclusively localizes unipolarly to the leading pole". Fig. 7 in Ref 25 would suggest that 30% of the cells have bipolar PilB localization. 70% is not "almost exclusively" unipolar. For PilT, the guesstimate from these data is that bipolar symmetric is about 10%, unipolar 20%, and bipolar asymmetric about 70%. It is therefore misleading to state that PilT localizes bipolarly.

Reviewer #2 (Remarks to the Author):

The authors have answered my questions.

Reviewer #3 (Remarks to the Author):

The authors have addressed most of my concerns regarding clarity of writing and experiments. However, I still do not see this work as novel, as there are several reports of developmental localization of c-di-GMP enzymes to cellular sites. The authors responded to this by saying that in those previous reports "either the relevant DGC and PDE localize to opposite cell poles or a PDE localizes unipolarly. To the best of our knowledge, our results provide the first example where c-di-GMP mediates the generation of equal daughter cells and this involves placing a DGC at mid-cell of the pre-divisional cell." There is in fact a report of a DGC localizing at the mid cell of a pre-divisional cell to inhibit division, that the authors appear to be unaware of. See PMID: 27507823 'A Diguanylate Cyclase Acts as a Cell Division Inhibitor'

Reviewer #1 (Remarks to the Author):

I regret to reiterate that my primary concerns persist. Firstly, there is apprehension regarding the selective and limited nature of the datasets pertaining to the cell division cycle. The wording (L196-203) suggests that the cells analyzed for DmxA localization (Fig. 3a) were chosen based on fluorescent signals. Additionally, for the localization of PilQ and RomR, about 30 cells in total were analyzed for both the WT and the mutant (Figs. 5a & 6a). Of particular concern is the former (DmxA) as it is pivotal to the model.

Response: L196-203 & Fig. 3a: In this experiment, we analyse in an unbiased manner 600 cells from three biological replicates with each 200 cells by snapshot analysis using fluorescence microscopy. In doing so, find that ~5% of all these 600 cells have a cluster at midcell, while the remaining cells have a speckled pattern. Looking closer, we find that the ~5% of cells with a midcell cluster are dividing cells but not all dividing cells have a cluster. Subsequently, we follow up on these observations and perform time-lapse microscopy on 100 cells starting at different stages of the cell cycle. In these experiments, we observe the DmxA-mVenus cluster forms at midcell late during division. Thus, we have two complementary and unbiased experimental approaches that both show that DmxA-mVenus localizes at midcell late during cell division. To further clarify, we modified the text in line 194-195 and 199-200. Moreover, we slightly reorganized the legend to Fig. 3a to make it even more clear how many cells were analysed in the experiments (line 1089, 1117).

Figs. 5a & 6a: In these two figures, we describe the outcome of unbiased fluorescence time-lapse microscopy experiments in which we follow the localization of PilQ-sfGFP and RomR-mCherry, respectively during cell division. In doing so, we follow 28/31 (PilQ-sfGFP) and 27/22 (RomR-mCherry) cell division events in WT and the $\Delta dmxA$ mutant. Subsequently, we go on to perform unbiased snapshot analyses on 1000s of cells by fluorescence microscopy in all four strains. Both analyses demonstrate that PilQ-sfGFP and RomR-mCherry localize aberrantly in the absence of DmxA. Moreover, the time-lapse microscopy experiments in Figs. 5a & 6a show that it is the distribution of the two proteins during cell division that is not working correctly in the absence of DmxA. Because we have two lines of evidence showing that PilQ-sfGFP and RomR-mCherry localisation is altered in the absence of DmxA, we feel confident that our conclusion is correct. We slightly modified the legend to the two figures to clarify (lines 1155 and 1183).

Second, the model's validity remains in question as the authors have not addressed concerns raised by multiple reviewers. More specifically, the points #2, #5, #6, #8, #11 and #12 raised by this reviewer are all concerns related to one aspect or another of the proposed model. The summary by Reviewer #2 concluded that the "reviewer is not convinced by the authors model and data interpretation". Essentially all major points raised by this reviewer focused on why this conclusion was reached. The review by Reviewer #3 stated that the "problematic part was the writing..., which over-sold and overinterpreted the findings". It further states that "the claim that localization mechanisms... is not grounded in reality. So also the claim that.....". These concerns should not be overlooked.

Response: We note that reviewer #2 was satisfied with our revised manuscript. We also note that reviewer #3 only commented on whether to include mentioning of YfiN (please see our response below).

The following are comments on their rebuttal to selected points this reviewer raised previously.
#1. None of the references indicated a stimulation of motility, only adhesion.

Response: We added Huang et al. 2003 (ref. 9) in which it is shown that FimX stimulates T4P-dependent motility. The Jain et al. 2017 reference (ref. 6) shows that c-di-GMP binding by FimX is important for its function in stimulating T4P formation. We added Varga et al 2006 (ref. 10) in which it is shown that *Clostridium perfringens* moves by T4P-dependent motility. In Hendrick et al. 2017 (ref. 8), it is shown that c-di-GMP binds the PilB extension ATPase in *C. perfringens* and stimulates its function in T4P extension. The two additional references were added in line 50.

#3. There is no calibration with MW standards of a prep SEC column. The elution may suggest a change in oligomeric state, but not dimer vs monomer or pentamer vs hexamer.

Response: We added the calibration curve for the SEC experiment in Fig. S1a. Moreover, we changed the text in line 113-115 to clarify that the elution profiles fit with dimers and monomers of globular proteins. Finally, we amended the Methods section in line 669-670.

#4. This goes back to a 2016 publication that measured c-di-GMP at pmol per mg protein. There is no explanation provided how this unit was converted to micromolar, which requires the conversion of protein to volume.

Response: In the Skotnicka et al. 2016 paper we express the c-di-GMP concentration in pmol/mg protein. In the Skotnicka et al. 2020 paper, we converted this number to the more relevant molar concentration of c-di-GMP in an average *M. xanthus* cell. We did this as follows: We know the concentration of c-di-GMP in our sample(s) and its total volume(s), therefore, we can calculate the total amount of c-di-GMP in the measured sample(s). We also know the number of cells that the sample was prepared from. Combining these two numbers we can calculate moles of c-di-GMP per cell. Using the average dimensions of a *M. xanthus* cell, we can then calculate the c-di-GMP concentration in an average *M. xanthus* cell.

#7. The binary statement on PilB & PilT (L318) localization appears misleading. That is, it is inaccurate to state that "PilT localizes bipolarly, while PilB almost exclusively localizes unipolarly to the leading pole". Fig. 7 in Ref 25 would suggest that 30% of the cells have bipolar PilB localization. 70% is not "almost exclusively" unipolar. For PilT, the guesstimate from these data is that bipolar symmetric is about 10%, unipolar 20%, and bipolar asymmetric about 70%. It is therefore misleading to state that PilT localizes bipolarly.

Response: We modified the text in line 319-321 to clarify.

Reviewer #2 (Remarks to the Author):

The authors have answered my questions.

Response: Thank you!

Reviewer #3 (Remarks to the Author):

The authors have addressed most of my concerns regarding clarity of writing and experiments. However, I still do not see this work as novel, as there are several reports of developmental localization of c-di-GMP enzymes to cellular sites. The authors responded to this by saying that in those previous reports "either the relevant DGC and PDE localize to opposite cell poles or a PDE localizes unipolarly. To the best of our knowledge, our results provide the first example where c-di-GMP mediates the generation of equal daughter cells and this involves placing a DGC at mid-cell of the pre-divisional cell." There is in fact a report of a DGC localizing at the mid

cell of a pre-divisional cell to inhibit division, that the authors appear to be unaware of. See PMID: 27507823 'A Diguanylate Cyclase Acts as a Cell Division Inhibitor'

Response: First of all, thank you for pointing us to this reference. In PMID: 27507823, the authors show that the diguanylate cyclase YfiN in *Escherichia coli* and *Salmonella enterica* localizes to the division site in response to redox stress and envelope stress; thus, this YfiN localization is not hard-wired into the cell cycle. At the division site, YfiN inhibits cell division. In our analyses, *M. xanthus* cells are not exposed to stress and, thus, the localization of DmxA to the division site is not a response to stress but hardwired into the cell cycle. Moreover, we have no evidence suggesting that DmxA causes inhibition of cell division. In our text (line 483-496) we specifically write that “Several bacteria that alternate between a planktonic, flagellum-dependent swimming lifestyle and a surface-associated lifestyle, harness c-di-GMP to deterministically generate phenotypically distinct daughters during division^{4, 11}. In *C. crescentus*, *P. aeruginosa* and *Shewanella putrefaciens*, the programs driving the generation of this heterogeneity rely on the asymmetric deployment of c-di-GMP metabolizing enzyme(s) to the daughters during cell division, i.e. either the relevant DGC and PDE localize to opposite cell poles or a PDE localizes unipolarly^{7, 45, 55, 66-72} (Fig. 7). Consequently, one daughter has low c-di-GMP and becomes the flagellated swimming daughter, while the other daughter has high c-di-GMP and becomes the surface-associated daughter. By contrast, *M. xanthus* places the DGC DmxA at the division site, thereby enabling the formation of mirror-symmetric and phenotypically similar daughters. Thus, by deploying c-di-GMP synthesizing and degrading enzymes to distinct subcellular locations, bacteria harness c-di-GMP to establish deterministic programs that are hardwired into the cell cycle to generate or, as shown here, minimize phenotypic heterogeneity (Fig. 7)”. Thus, we do not state that DmxA is the first DGC to be shown to localize to the division site, but that this localization mediates the formation of mirror-symmetric and phenotypically similar daughters.

C-di-GMP is involved in a variety of different processes in bacteria that we cannot all cover. Therefore, we would like to keep the focus of the discussion on the role of c-di-GMP in establishing deterministic programs that are hardwired into the cell cycle to generate or, as shown here, minimize phenotypic heterogeneity. Therefore, we decided not to include a discussion of the findings reported in PMID: 27507823.